# Size-resolved aerosol and cloud condensation nuclei (CCN) properties in the remote marine South China Sea, Part 1: Observations and source classification

Samuel A. Atwood[1], Jeffrey S. Reid[2], Sonia M. Kreidenweis[1], Donald R. Blake[3], Haflidi H. Jonsson[4], Nofel D. Lagrosas[5], Peng Xian[6], Elizabeth A. Reid[2], Walter R. Sessions[6,7], James B. Simpas[5]

[1]Department of Atmospheric Science, Colorado State University, Ft. Collins, CO
[2]Marine Meteorology Division, Naval Research Laboratory, Monterey, CA
[3]Department of Chemistry, University of California, Irvine, CA
[4]Department of Meteorology, Naval Postgraduate School, Monterey, CA
[5]Manila Observatory, Manila, Philippines
[6]CSC Inc. at Naval Research Laboratory, Monterey, CA
[7]Space Sciences Engineering Center, University of Wisconsin, Madison, WI

*Correspondence to*: Jeffrey S. Reid (jeffrey.reid@nrlmry.navy.mil)

**Abstract.** Ship-based measurements of aerosol and cloud condensation nuclei (CCN) properties are presented for two weeks of observations in remote marine regions of the South China Sea/East Sea during the Southwestern Monsoon (SWM) season. Smoke from extensive biomass burning throughout the Maritime Continent advected into this region during the SWM, where it was mixed with anthropogenic continental pollution and emissions from heavy shipping activities. Eight aerosol types were identified using a K-Means cluster analysis with data from a size-resolved CCN characterization system. Interpretation of the clusters was supplemented by additional onboard aerosol and meteorological measurements, satellite, and model products for the region. A typical bimodal marine boundary layer background aerosol population was identified and observed mixing with accumulation mode aerosol from other sources, primarily smoke from fires in Borneo and Sumatra. Hygroscopicity was assessed using the $\kappa$ parameter and was found to average 0.40 for samples dominated by aged, accumulation mode smoke; 0.65 for accumulation mode marine aerosol; 0.60 in an anthropogenic aerosol plume; and 0.22 during a short period that was characterized by elevated levels of volatile organic compounds not associated with biomass burning impacts. As a special subset of the background marine aerosol, clean air masses substantially scrubbed of particles were observed following heavy precipitation or the passage of squall lines, with changes in observed aerosol properties occurring on the order of minutes. Average CN number concentrations, size distributions, and $\kappa$ values are reported for each population type, along with CCN number concentrations for particles that activated at supersaturations between 0.14% and 0.85%.

**Keywords.** Remote Marine Aerosol, Cloud Condensation Nuclei, Biomass Burning, South China Sea, Source Apportionment, K-Means, Mixing State

## 1 Introduction

In the Southeast Asian Maritime Continent (MC) and South China Sea/East Sea (SCS) aerosol particles are expected to play an important role modulating cloud development, precipitation, and radiative properties that affect heat transfer through the atmosphere (Reid et al., 2013). Assessment of aerosol properties important to understanding such processes in remote marine segments of this region has proven difficult. Extensive cloud cover confounds remote sensing and leads to a clear sky bias in observations (Feng and Christopher, 2013; Reid et al., 2013). Aerosol monitoring has largely been confined to urban centers that are often dominated by local emissions, while in-situ sampling in remote areas has been limited in duration and scope (Irwin et al., 2011; Robinson et al., 2011; Lin et al., 2014; Reid et al., 2015). Airborne measurements have provided some representation of aerosol over wider regions and at various levels (Hewitt et al., 2010; Robinson et al., 2012), but additional questions regarding the representativeness of such point measurements across larger time scales remain. Similarly, the impact of various aerosol sources on surface properties and concentrations in remote marine regions, and their relationship to expected transport pathways and the few remotely sensed column measurements that exist is not well understood. Thus, over these remote ocean regions the aerosol optical and physical properties, their variability in time and space, and the processes controlling aerosol lifecycle have not been well constrained. This uncertainty in the aerosol environment itself comes in addition to uncertainty about its impacts on meteorological processes. Aerosol concentration has been found to relate to cloud development, cloud microphysics, and precipitation formation in the region (Yu et al., 2008; Yuan et al., 2011; Wang et al., 2013), while smoke may affect cloud droplet size distributions and the onset of precipitation, similar to processes observed in other tropical regions impacted by biomass burning (Rosenfeld, 1999; Andreae et al., 2004). Improved knowledge of the aerosol environment and aerosol-cloud-climate relationships in the Southeast Asian region has therefore been identified as important regionally, and in regards to links with global climate and large-scale aerosol budgets (Reid et al., 2013).

During the May through October Southwestern Monsoon (SWM) season, burning throughout the MC typically reaches its greatest extent between August and early October as precipitation associated with the ITCZ shifts north into Indochina (Reid et al., 2012). The resulting heavy smoke mixes with urban, industrial, marine, and shipping emissions in an exceedingly complex aerosol mixture (Balasubramanian et al., 2003; Atwood et al., 2013; Reid et al., 2013). During this period, aerosol particles from surface sources are generally advected by low level mean winds throughout the SCS, where they are scavenged by precipitation or eventually removed in the monsoonal trough east of the Philippines (Reid et al., 2012, 2015; Wang et al., 2013; Xian et al., 2013). As a result, the region of the SCS and Sulu Sea to the north and east of Borneo has been predicted to be a receptor for much of these biomass burning and pollution emissions from the greater MC during periods when air masses enter more convective phases of the SWM (Reid et al., 2012; Xian et al., 2013).

Remote marine aerosol and its impact on atmospheric processes have been studied in a number of ocean regions (Hoppel et al., 1986; Russell et al., 1994; Jensen et al., 1996; Brechtel et al., 1998; Murphy et al., 1998; Bates et al., 2000; Petters et al., 2006; Quinn et al., 2006). These studies identified a background submicron marine aerosol that is composed of two distinct

modes in the number distribution, due to processing by non-precipitating clouds (Hoppel et al., 1986, 1994; Hudson et al., 2015). Bates et al., (2000) linked the differences in the average size distributions of background marine aerosol in two remote marine regions to regional meteorology, including differences in aerosol residence time and cloud processing. Increased wind speeds lead to increased flux of sea-salt particles into the atmosphere, contributing submicron particles as small as 40 nm in diameter (O'Dowd and Leeuw, 2007; Russell et al., 2010; de Leeuw et al., 2011; Bates et al., 2012; Modini et al., 2015). Non-seasalt-sulfate and organic matter from marine sources also comprise large fractions of the submicron aerosol mass loading in clean and background marine air masses (Murphy et al., 1998; Cavalli et al., 2004). As air masses from more terrestrial or anthropogenically-influenced regions advect over remote marine regions, submicron size distributions and chemical compositions often diverge from background conditions (Bates et al., 2000; Quinn et al., 2006). More recent studies have further quantified the role of various processes in shaping the marine aerosol population, including primary and secondary production, aging, and mixing with non-marine sources (Allan et al., 2009; Russell et al., 2010; de Leeuw et al., 2011; Bates et al., 2012; Prather et al., 2013; Frossard et al., 2014; Modini et al., 2015). In particular, the contribution of dissolved organic components in the sea surface micro layer to aerosol produced by bubble breaking has been noted, with increasing organic enrichment as size decreases (Russell et al., 2010; Bates et al., 2012; Prather et al., 2013; Quinn et al., 2014). Additional studies into the source dependent composition of marine aerosol have indicated non-marine sources can be important contributors to aerosol in marine regions. Shank et al., (2012) found evidence of biomass burning and combustion impacts on remote marine MBL aerosol, including in nominally clean marine conditions. These authors also noted the limited importance of organic components in particulate matter in a tropical Pacific location, as compared to other regions where organics were a more important fraction of the submicron aerosol. Frossard et al., (2014) found influences on aerosol organic matter from shipping and mixing with non-marine sources in 63% of observations across five ocean regions. Modini et al., (2015) evaluated the contribution of primary marine aerosol to cloud condensation nuclei (CCN) number concentrations, and found that it accounted for less than 10% of CCN active at 0.9% supersaturation during low wind conditions, with increasing importance (up to 58% of CCN) at higher wind speeds and lower environmental supersaturations. Taken as a whole, recent understanding of marine aerosol indicates that the background marine aerosol and primary marine emissions can be complex and play an important role in cloud, radiative, and precipitation processes, and that other sources of aerosol contribute to number and mass concentrations, even in relatively clean and/or remote regions.

Two research cruises were conducted in the remote marine boundary layer (MBL) of the SCS and Sulu Sea during the 2011 and 2012 SWM seasons to perform in situ aerosol and meteorological measurements, and to investigate marine aerosol and its impacts on clouds, precipitation, and climate as it reflects the complex set of sources in the region (Reid et al., 2015, 2016). In this paper, we present observations of aerosol and CCN characteristics during the second cruise, along with their relationship to aerosol source type, air mass, and meteorological phenomena. These measurements represent the first in situ observations of size-resolved CCN properties in the area, and fill a gap in knowledge needed to assess aerosol-cloud-precipitation relationships in the data poor remote marine SCS region.

## 2 Methods and Cruise Description

The SCS research cruises occurred during the month of September in both 2011 and 2012, and took place aboard the 35m, 186 ton *M/Y Vasco*, operated out of Manila by Cosmix Underwater Research Ltd. A thorough description of the vessel and instrumentation for the 2011 and 2012 cruises can be found in Reid et al. (2015) and Reid et al. (2016), respectively. Here
we are concerned with the 2012 cruise, which departed Manila on 4 September from Navotas, Manila Bay, and returned on 29 September. The sampling strategy for these cruises involved moving between various anchorages in the SCS and Sulu Sea around the Palawan Archipelago. Sampling occurred throughout the cruise, but aerosol measurements were shut down or invalidated and removed from the dataset during periods when representative sampling could not be achieved (i.e., measured relative wind not from over the ship bow, leading to potential self-sampling; see Reid et al., (2016) for more details).
Remaining periods of self-sampling of ship emissions were identified by anomalous size distributions and high particle number concentrations, and were removed from the data set before analysis. The size-resolved CCN system (Petters et al., 2009) that provided the bulk of the measurements reported here had a computer failure that, once fixed, limited reliable measurements to primarily the last two weeks of the cruise, hence we focus here on data from 14-26 September 2012. This period included several transits along the east side of Palawan Island, stationary measurements at an anchorage between
Palawan and Borneo (7.86N, 116.94E), and two anchorages at Tubbataha Reef in the middle of the Sulu Sea (8.80N, 119.26E).

### 2.1 Aerosol Measurements

A DMT Passive Cavity Aerosol Spectrometer Probe (PCASP) X2 configured in an aviation pod with heated inlet was mounted at the *Vasco* top mast approximately 10 m above the water surface to provide optical measurements of dry particle
size distributions between approximately 125 nm and 3 μm. Additional aerosol instrumentation was located in a forward locker, below and slightly behind the aerosol inlet. Sampled air was fed to the locker via a 3 cm diameter, 4 m long inlet located next to the PCASP. Although the inlet was not aspirated, several high flow rate nephelometers sampling from the inlet ensured low residence time in the sample line. Further details and additional instrumentation are discussed in Reid et al. (2016). A size resolved CCN system sampled air from the inlet just inside the instrument locker with an approximately 2 m,
0.635 cm diameter copper tube. A URG cyclone with an approximate 1 μm 50% size cut was used to remove the largest particles from this CCN sampling line, including coarse mode sea-salt, to minimize wear and corrosion on downstream components. The sample was then dried using a Permapure poly-tube Nafion dryer with low pressure sheath air to RH values below 30%, as verified by occasional in line checks using a handheld Extech Hygro-Thermometer.

Approximately 1.1 liters per minute (lpm) were drawn through the size resolved CCN system, which was comprised of an x-
ray neutralizer (TSI Model 3087) and a TSI 3080 Electrostatic Classifier with a long DMA column (TSI Model 3081) for quasi-mono-disperse particle sizing, preceded by a 0.071 cm orifice (0.69 μm 50% cutpoint diameter) impactor. The DMA was operated with a sheath flow rate of 5 lpm (liters per minute) and sample flow rate of 1.1 lpm. The sample flow was then

split between a TSI 3782 Water Condensation Particle Counter (CPC) with a flow rate of 0.6 lpm, and a DMT Cloud Condensation Nuclei Counter (CCNc) with a flow rate of 0.5 lpm. Flow rates used to calculate number concentrations were calibrated using a Gilibrator (Models 800285 & 800286) system, with in-line measurements conducted prior to each CCNc supersaturation calibration session as noted below.

The size-resolved CCN system measured CN and CCN (activated particles at a CCNc set-point supersaturation) concentrations in each of 30 quasi-mono-disperse size bins between 17 nm and 500 nm. The CCNc was operated at five temperature gradient settings and calibrated using ammonium sulfate (following the methods described by Petters et al. (2009)) to measure the corresponding maximum environmental supersaturation within the CCNc column. The scan of all 30 size bins at each supersaturation took approximately 15 minutes, while a complete measurement over all 5 supersaturation

settings took approximately two hours due to pauses between settings while column temperatures stabilized. The measured CN and CCN particle counts were inverted using the methodology of Petters et al. (2009). The inversion yielded the dry ambient aerosol size distribution over the measured range ($dN/dlog_{10}D_p$ for $17 \leq D_p \leq 500$ nm) and the equivalent distribution of CCN particles activated at each supersaturation ($dCCN/dlog_{10}D_p$). The activated fraction spectrum was then calculated as the fraction of total particles that formed droplets (CCN/CN) at each diameter. Each activated fraction spectrum

was then fit using a three parameter fit similar to the approach of Rose et al. (2010). The diameter at which 50% of particles in the fit had activated ($D_{50}$) was used to calculate the associated hygroscopicity parameter, $\kappa$ (Petters and Kreidenweis, 2007). Full calibration of the CCN system flow rates and supersaturations took two to five hours to complete, and was therefore conducted four times throughout the study on 15, 20, 27 and 29 September to limit measurement downtime. Each calibration session involved running a calibration scan at each CCNc temperature gradient setting (with two full scans

conducted at each setting on 27 Sep) yielding a total of five calibrations per setting throughout the cruise. Calibrated supersaturation set-points and their respective standard deviations were 0.14% ± 0.01, 0.38% ± 0.01, 0.52% ± 0.01, 0.71% ± 0.02, and 0.85% ± 0.03, with no significant trend or calibration drift noted during the cruise. The measured range of 0.14% to 0.85% supersaturation was selected based on values that could both be reliably measured by the CCNc instrument and represented supersaturations expected in the region where aerosol effects may be relevant, ranging from marine

stratocumulus with peak supersaturations often below 0.2% to highly convective clouds with supersaturations above 1% (Reutter et al., 2009; Ward et al., 2010; Tao et al., 2012).

As the SCS environment tended to have relatively few particles smaller than 50 nm, only the measurements at the 0.14% and 0.38% supersaturation settings had complete activation curves that spanned the measured particle diameter range. For the higher supersaturation settings, the $D_{50}$ diameter tended to occur at small diameters with low CN and CCN concentrations,

thereby increasing uncertainty in the associated $\kappa$ values. As a result, $\kappa$ values were only reported for the 0.14% and 0.38% supersaturation settings. In addition, rather than continuously running a full two-hour scan across all supersaturation settings, individual scans (approximately 15 minutes) were run more often for the 0.14% and 0.38% settings to take advantage of this outcome. The range of $\kappa$ values measured for the particles active at supersaturations of 0.14% and 0.38% was typically

between about 0.3 and 0.8, although the full range was between 0.2 and 1.1 (Figures 2e, f). The $D_{50}$ diameters for these hygroscopicity values spanned approximately 96 to 150 nm (κ range: 0.8 – 0.2) for the 0.14% supersaturation setting, and 45 to 80 nm (κ range: 1.1 – 0.2) for the 0.38% setting. The hygroscopicity measurements were therefore more indicative of accumulation mode hygroscopicity during the 0.14% scans and Aitken mode hygroscopicity during those at 0.38%.

Additionally, particles outside these size ranges were not well quantified by these measurements.

## 2.2 Ancillary Measurements and Products

Additional measurements of aerosol composition were used to validate identified source types impacting the measurements throughout the cruise. A series of $PM_{2.5}$ filters were collected by 5 lpm Minivol Tactical Air Samplers with sampling periods that varied between one and two and half days, and were analyzed for elemental concentrations by gravimetric, XRF, and ion

chromatography methods, and organic and black carbon concentrations by the thermal-optical methods (Reid et al., 2016). Trace gas concentrations were measured intermittently throughout the cruise by whole-air gas samples for gas chromatography analysis, as discussed further in Reid et al. (2016).

A suite of weather monitoring instruments was located on a 3m bow mast to provide coincident meteorological measurements throughout the study. From this suite, wind speed and wind direction measurements from a Campbell sonic

anemometer were used to identify gust front passage. An OTT Parsivel disdrometer was utilized to measure precipitation, from which only the rain rate measurements were used in this analysis.

Several remote sensing and model products were used to characterize the wider SCS atmospheric environment and to identify potential aerosol sources. MODIS visible and IR products were used to identify convection and squall line propagation across the SCS. The MODIS Collection 6 MOD08 Level 3 daily Aerosol Optical Depth products were utilized

for AOD measurements in the region, though cloud cover obscured measurements throughout much of the study. MODIS active fire hotspot analysis and the FLAMBE smoke flux product from Terra and Aqua were used to identify the locations and times during which fires were burning in the MC (Giglio et al., 2003; Reid et al., 2009; Hyer et al., 2013). Simulations from the Navy NOGAPS model were used to represent surface and 700 hPa winds, interpolated to one-degree spatial resolution and averaged over the study period, to provide an estimate of typical aerosol transport pathways (Hogan and

Rosmond, 1991; Xian et al., 2013). Finally, the Navy Aerosol Analysis and Prediction System (NAAPS) was used to predict smoke and sulfate aerosol mass concentrations at the surface along the *Vasco* ship track (Lynch et al., 2016).

The NOAA Hybrid Single Particle Lagrangian Integrated Trajectory (HYSPLIT) Version 4.9 model (Draxler et al., 1999; Draxler and Hess, 1997, 1998) was used to generate daily 72-hour backtrajectories (spawned at 0Z, 8 AM local) from the *Vasco* location with arrival heights of 500m to indicate likely marine boundary layer transport patterns. The GDAS1, 1° x 1°

HYSPLIT meteorological dataset was used to drive the model. Trajectory paths were found to be largely influenced by synoptic scale changes in the regional meteorological state of the atmosphere, with no substantial differences due to arrival time of day. Arrival heights between 100 m and 3000 m were examined. Trajectories with arrival heights below 1000 m were generally consistent and representative of boundary layer transport (Atwood et al., 2013; Xian et al., 2013), while

higher heights tended to be increasingly influenced by free troposphere transport pathways with a more westerly component. As such, 500 m was selected to be representative of general shifts in synoptic scale boundary layer transport pathways, though more complex vertical interactions and mixing from aloft are a potential influence in the region (Atwood et al., 2013). Trajectory lengths of 72 hours were found to be sufficient to demonstrate general transport path differences between ocean dominated regions of the central portion of the SCS and more terrestrially influenced regions that passed closer to Borneo and Sumatra.

## 2.3 Aerosol Population Type Classification

The $dN/dlog_{10}D_p$ dry particle size distributions obtained every 15 minutes from the CCN system were first normalized by the particle number concentration summed over all bins. Each of these normalized size distributions was then parameterized by fitting a lognormal mixture distribution using an algorithm based on Hussein et al., (2005). This algorithm fit each distribution to one, two, or three lognormal modes, each defined by three lognormal distribution parameters (median diameter, geometric standard deviation, and fractional number concentration). Two modes were identified as the best fit for 690 of 731 data points in this dataset. In order to have a common point of comparison for classification purposes the two mode fit was used for all data points, yielding a parameterized Aitken and accumulation mode for each 15-minute data point. Data points for which the fitting algorithm selected an additional third mode are noted in the clustering results in Sect. 3 and Supplementary Figure S1.

Parameters associated with each data point were then used to cluster the data points into groups based on similar observed aerosol properties. Cluster analyses have long been used to group observed aerosol size distributions into clusters of generally similar size distributions (Tunved et al., 2004), which can then be associated with various sources or atmospheric processes that shaped them (Charron et al., 2008; Beddows et al., 2009; Wegner et al., 2012). Similar cluster analyses have been utilized to classify aerosol types based on particle chemistry (Frossard et al., 2014), with Frossard et al., (2014) identifying clusters in marine aerosol observations associated with marine and non-marine aerosol types. As pointed out by those authors, the clustering approach can be superior to algorithms using simpler criteria to distinguish "clean" from "polluted" conditions, as more variables and a measure of similarity between data points are used to find the underlying population types. In this study normalized size distribution parameters were combined with total number concentration and particle composition information via hygroscopicity measurements to serve as input variables for the cluster analysis (parameters given in Table 1). As hygroscopicity data were available for at most one mode (during the 0.14% or 0.38% supersaturation scans), Aitken and accumulation mode hygroscopicities were treated as missing for data points without this information. In order to account for missing data and adjust all clustering variables to the same scale, each variable was first standardized to a mean of zero and standard deviation of one, with missing data points imputed to a value a zero (the mean value). As a result, the clustering distance function was insensitive to missing data, but still included information on hygroscopicity when available.

A hierarchical cluster analysis was first conducted using the *cluster.AgglomerativeClustering* class of the Python scikit-learn package (Pedregosa et al., 2011) using the Ward linkage to help ascertain the number of clusters that can be found in the dataset. Each step in this process involved merging two data points or clusters into a new cluster based on those points with the shortest distance between normalized input measurements (Karl Pearson Euclidian distance function; Wilks, 2011). A dendrogram and associated measure of the distance between merged clusters for each subsequent clustering step was used to identify potential numbers of clusters appropriate for the dataset. The distance between merged clusters increases at steps that merge substantially different clusters (Wilks, 2011), in this case indicating 5, 8, 9, and 12 clusters as potentially appropriate for this data set.

A nonhierarchical K-Means cluster analysis was then conducted for each of these four potential cluster numbers using the scikit-learn *cluster.KMeans* class to refine the cluster members. The appropriate number of clusters was selected based on the K-Means result with the least number of clusters that maintained physically distinct and temporally consistent aerosol populations for the associated clusters. In particular, as the timestamp of a data point was not included in the cluster analysis, clusters with smaller numbers of data points were considered distinct if they all occurred during a narrow time frame that could be associated with a transient atmospheric phenomenon. The time frames of all clusters were then compared against other aerosol and meteorological observations to ensure they were physically meaningful. The result was a set of eight identified aerosol population types with associated time periods corresponding to the 15 minute CCN system data points. Finally, size distribution and hygroscopicity measurements were averaged for all time periods associated with each population type.

## 3 Results

### 3.1 Overview of Study

The daily positions of the *Vasco* during the two weeks in September 2012 that comprise this study are shown in Figure 1a, together with HYSPLIT 72-hour backtrajectories initiated within the MBL. Several extended periods at the same anchorage are noted by the range of dates spent at these locations. The daily fire hot spots are also indicated in this figure. Average NOGAPS surface winds in the boundary layer were from the southwest, often advecting air parcels from near Borneo, while lower free tropospheric winds, such as those at 700 hPa, were more westerly due to the generally veering structure of the lower atmosphere in the SCS (Reid et al., 2016). Fires detected by MODIS occurred throughout much of Borneo and Southern Sumatra during the study, with surface-level trajectories near the start and near the end of the study period passing close to active fires, whereas those during the middle period remained primarily over open ocean. Results from the NAAPS model, along with limited satellite AOD measurements not obscured by clouds during this period, confirmed this general smoke transport pathway. Accumulation mode aerosol mass concentration estimates (Figure 1b) were initially generated from the PCASP measurements using a density of 1.4 µg m$^{-3}$ (Levin et al., 2010), assumed to be representative of a combination of smoldering peat and agricultural fire emissions typical in the MC (Reid et al., 2012) that constituted the

largest plumes observed during the study (Reid et al., 2016). Coincident model estimates generated by NAAPS along the ship track indicated generally similar results, with the highest mass concentrations occurring early and late in the measurement period, in general agreement with times during which backtrajectories passed over terrestrial sources and active fires. Air parcels advecting into the SCS and Sulu Sea during this period that originated from areas further to the north and west were cleaner than those from other sectors due to fewer emission sources and more precipitation along the trajectories. Changes in the observed particle concentrations also occurred on timescales shorter than these weekly large-scale variations, as shown in the aerosol observation timelines in Figures 2a-c. Many of these higher-frequency fluctuations were associated with squall line passages and heavy local precipitation, as discussed further below. The timeline of dN/dlogDp size distributions, as measured by the CCN system and shown in Figure 2a, indicates that most of the particle number concentration fell within the 17-500 nm measurement range, except possibly during the highest-concentration periods. A more extensive comparison of model and satellite measurement in situ observations is discussed in Reid et al. (2016).

## 3.2 Aerosol Population Type Classification and Properties

The cluster analysis was first conducted to investigate potential aerosol population types in the dataset, followed by physical interpretation of the results against cluster aerosol properties, coincident measurements, and meteorological conditions. The parameter values input to the cluster analysis are shown for each data point and variable in Figure 3, and colored by cluster number for the results of the eight-cluster K-Means analysis. The average value and intra-cluster standard deviation for each cluster parameter and cluster are given in Table 1. Normalized size distributions for each of these eight aerosol populations are shown in Figure 4; the average CN and CCN number concentrations and hygroscopicities are given in Table 2. Equivalent normalized volume distributions are shown in Supplementary Figure S2. The cluster number associated with each measurement is similarly shown as the background color in Figure 2 and marker color in Figure 5. The aerosol properties, meteorological conditions, and likely transport pathways associated with data points in each cluster were then used to provide a physical interpretation of the results and identify each population type on the basis of its likely sources as discussed below. Clusters 1-4 were the most commonly found (representing 85% of the total observations, Table 2), while clusters 5-8 represented special cases, generally of short duration, that could be identified by specific locations or sampling conditions.

1.  Background Marine: Data points associated with this cluster occurred throughout the study, typically following rain in the vicinity of the *Vasco* or transport from areas further removed from terrestrial regions. In addition, this type was observed following shortly after periods associated with each of the other identified clusters, often appearing as a transition between other types (Figure 2). The measured properties of this population type were similar to the background marine aerosol reported in many prior studies (Hoppel et al., 1994; Jensen et al., 1996; Brechtel et al., 1998; O'Dowd et al., 1997; Heintzenberg et al., 2004; Allan et al., 2009; Good et al., 2010). The population featured a bimodal size distribution with a Hoppel minimum near 90 nm. The inner quartile range (IQR: middle

50% of observations between the 25%−75% percentiles) of number concentrations ranged from 382 to 623 cm$^{-3}$, with on average 42% of the total number concentration residing in the accumulation mode as specified by the bimodal fit. Modal hygroscopicities were found to average 0.65 for the accumulation mode and 0.46 for the smaller Aitken mode, while activated fractions were generally moderate across the range of measured supersaturations as compared to other identified population types (Table 1). Each of these findings further reinforced the classification of this cluster as a typical background marine aerosol.

2. Precipitation: This distribution was found during periods immediately following extensive precipitation at or near the *Vasco* (Figure 2d). Air masses had been substantially scrubbed of particles and accumulation mode particles had been preferentially removed. While the number concentrations of large-mode particles were lower than those in the background marine periods, the number concentrations of smaller particles, particularly those below 40-50 nm, were comparable to the background marine type. The longest contiguous period of this type occurred on 14 Sept. immediately following the passage of a squall line observed in the satellite visible and IR products (not shown) that left a clean air mass with fewer than 200 cm$^{-3}$ measured in the 17-500 nm range in its wake. Number concentrations tended to be lower than the background marine type with an IQR from 227 to 441 cm$^{-3}$. Hygroscopicities were similarly lower than the background marine population with κ values of 0.54 and 0.34 for the accumulation and Aitken modes, respectively. Total CCN concentrations across all supersaturations were found to be lower than in background marine air masses due to the combination of fewer total particles, generally smaller particle sizes, and lower hygroscopicities.

3. Smoke: Data points associated with this aerosol type occurred primarily in two events on 14 Sept. and 25-26 Sept., during which backtrajectories were at their furthest south, near burning regions in Borneo (Figure 1a). Normalized size distributions indicated that particles were largely concentrated in a single accumulation mode with a tail of smaller particles. This type was associated with the highest total particle number and estimated submicron mass concentrations observed during the cruise, with the exception of measurements taken in the urban plume of Puerto Princessa. The standard deviations in the normalized size distribution parameters for the dominant accumulation mode in this population (Figure 4, Table 1) were small, even while number concentration varied widely (IQR 1802 to 2780 cm$^{-3}$; 81% accumulation modal fraction). Accumulation mode hygroscopicities were lower than either the background marine or precipitation types, with average κ values of 0.40. Aitken mode hygroscopicities showed the opposite behavior from the first two population types with higher κ values of 0.54, though the measured uncertainties (Figure 2 e&f) and standard deviations were considerably higher than for the accumulation mode (0.25 and 0.03, respectively). Interestingly, activated fractions were highest among all population types across the full range of measured supersaturations, owing to the large number fraction of particles in the accumulation mode, while CCN concentrations were the highest of all types (except those measured in port) due primarily to the larger total particle number concentration in these smoke plumes.

Accumulation mode lognormal median diameters around 200 nm with a tail of smaller particles, elevated concentrations of carbon monoxide and benzene, as well as potassium in filter samples during this period (Reid et al., 2016, and Figure 2c) were all consistent with expectations for aged biomass burning smoke (Yokelson et al., 2008; Akagi et al., 2011; Reid et al., 2015; Sakamoto et al., 2015). Additional examination and attribution of this event to biomass burning in Sumatra and Borneo is discussed further in Reid et al., (2016). Finally, while smoke is considered the dominant aerosol source during these periods, anthropogenic pollution may still have been co-emitted along the transport path and contributed to measured results.

4. Mixed Marine: This population was characterized by periods during which the background marine type mixed with other sources of aerosol. Most of the data points associated with this type had transport pathways and biomass burning sources similar to those for the smoke population type, but with number concentrations and size distribution parameters between those of the background marine and smoke types (IQR 782 to 1160 cm$^{-3}$; 68% accumulation modal fraction), indicating there was insufficient smoke for it to dominate the properties of the marine background. Accumulation and Aitken mode hygroscopicities of 0.48 and 0.54, along with activated fractions and CCN concentrations, were similarly indicative of mixing between smoke and background marine sources.

While periods of smoke mixing with a background marine air mass appeared to constitute the majority of data points in this cluster, several other periods point to other phenomena of interest being included in this type, perhaps indicating this cluster was relatively more complex than other population types. Short lived intrusions (two to five hours) of accumulation mode particles were regularly observed in both the CCN system and PCASP datasets (e.g. 18-23Z on 22, 23, and 24 Sep) after which the size distributions quickly returned to background marine conditions. These excursions were largely constrained to the pre-dawn hours (sunrise occurs around 22Z) when the boundary layer was thinnest, and when precipitation was occurring in the vicinity of the *Vasco*. Several prior studies have shown that smoke and anthropogenic pollution aerosol within the wider MC region can be lofted into and transported in the lower free troposphere (Tosca et al., 2011; Robinson et al., 2012; Zender et al., 2012; Campbell et al., 2013; Atwood et al., 2013). The influence of a free tropospheric aerosol layer as a source of MBL aerosol and CCN has been identified in other remote oceanic regions as well (Clarke et al., 2013). One possible explanation for these events (and possibly for the observed organic and ultrafine events that were characterized by increases in gas phase VOCs as noted in the next clusters) is therefore that aerosol may have been mixed down into the MBL from a layer aloft, perhaps on the edge of rain shafts. Alternatively, they may also be due to intermittent plumes of aerosol that survived stochastic precipitation removal events along a boundary layer transport pathway or human terrestrial activities in the pre-dawn hours. In addition, air masses influenced by anthropogenic pollution may have been included in this cluster as well, but without sufficiently different impacts on aerosol parameters to result in a distinct cluster.

5. Organic Event: An approximately four-hour period starting at 1Z on 23 Sept. had measured particle concentrations between 200 and 325 cm$^{-3}$, but with significantly ($p<0.001$) larger median diameters than either the precipitation or

background marine types (Figure 3). Both Aitken and accumulation mode particles had among the lowest hygroscopicities measured during the cruise, with κ values around 0.2. During this event measured concentrations of numerous VOCs were much higher than in gas canisters collected approximately 6 hours before and after it, with no associated increase in carbon monoxide (Reid et al., 2016; Figure 2c). The particles had lower hygroscopicities and larger sizes than the background marine particles observed just before this event. While the source of this event is uncertain, Robinson et al. (2012) found occasional organic aerosol above the boundary layer they attributed to biogenic Secondary Organic Aerosol (SOA) formation during an airborne campaign in the outflow regions of Borneo, while Irwin et al. (2011) reported κ values between 0.05 and 0.37 in a terrestrial, biogenically dominated MC environment. Such a source would be consistent with the observed population, perhaps due to growth of a background marine population by condensation of organics, although we lack the ancillary data needed to establish this.

6. Ultrafine Event: This cluster was associated with an approximately 20-hour period on 17-18 Sept. that included the highest concentration of particles below about 30 nm observed throughout the study (Figure 2a), and coincided with a period of elevated VOC measurements at the start of this event (Figure 2c). A filter during this period showed very low potassium concentrations, while benzene was among the lowest values measured during the study, indicating that biomass burning was not the likely source for this event. Anthropogenic, shipping, and marine and terrestrial biogenic emissions are known sources of such compounds; isoprene, a common biogenic VOC, was not observed during this event, and a brief period of elevated dimethyl sulfide, associated with marine emissions from phytoplankton, was observed shortly before—but not during—this event (Reid et al., 2016).

A tri-modal best-fit was indicated by the Hussein, et al. (2005) algorithm for a number of these data points (Figure 2a and Supplementary Figure S1). The period had an overall IQR of 482 to 661 cm$^{-3}$, with generally higher ultrafine number concentrations than other periods with similar total concentrations. The accumulation mode was similar in both size and hygroscopicity (κ = 0.65) to the accumulation mode of the background marine type, while the smaller Aitken mode showed larger modal fractions and overall number concentrations, and slightly higher hygroscopicities (κ = 0.50) as compared to the background marine measurements. However, we note that the 0.38% supersaturation hygroscopicity measurement would likely not have been sensitive to these below 30 nm particles, and therefore was likely not representative of this smallest third mode. Additionally, while total number concentration was slightly higher than the background marine population, measured CCN concentrations and activated fractions were generally lower, indicating many of the additional particles would not be expected to influence CCN concentrations until higher environmental supersaturations were reached. While not enough information is available to verify the nature of differences between ultrafine particles in these types, the results are consistent with an influx of smaller particles and VOCs into a background marine air mass, and were sufficiently distinct to be identified as a coherent period by the unsupervised K-Means analysis.

7. Transit: This type was associated with measurements taken during a transit away from the port of Puerto Princesa, a city with a population of over 200,000. During this period light, westerly winds advected anthropogenic pollution out over the Sulu Sea and along the path of the *Vasco*, allowing for sampling of the urban plume as it diluted and mixed with aerosol from other sources. Size distributions were dominated by an Aitken mode with a number median diameter around 80-90 nm, unique in measurements from this study, mixed with an accumulation mode with a smaller modal fraction than other types. The population had an IQR of 738 to 1029 $cm^{-3}$, while the generally decreasing number concentrations were consistent with an urban plume diluting and mixing with other aerosol populations. Modal hygroscopicity values of 0.58 and 0.62 for the accumulation and Aitken modes, respectively, were closer than those most of the other population types and consistent with high levels of sulfate aerosol in typical urban plumes.

8. Port: This type was assigned to the measurements taken during a short period in the port of Puerto Princesa. Local anthropogenic emissions were dominant during this period, with number concentrations that fluctuated between 4000 and 10,000 $cm^{-3}$. Ultrafine particles ($D_p$ < 100 nm) dominated number concentrations during this period, although large number concentrations of accumulation mode particles with diameters between 100 and 300 nm were also observed. As measurements were fluctuating rapidly and only one CCN scan at each supersaturation setting could be completed before instrumentation was shut down, hygroscopicity results were inconclusive and uncertain. This type is considered separate from the other types as it was not measured in a remote marine area away from the immediate influence of a nearby terrestrial source.

Finally, throughout the study coarse mode particles with diameters larger than about 800 nm were consistently observed in the PCASP volume distributions (Figure 2b). Concentrations of particles in this size range increased with increasing wind speed (Figure 5), consistent with generation of sea spray aerosol due to bubble breaking and wave action (O'Dowd and Leeuw, 2007). While the total number concentration of coarse particles is small compared to typical CCN concentrations (Figures 2e, f), in the cleanest conditions we measured, they represented non-trivial fractions of CCN active at 0.14% and 0.38% supersaturations. The large diameter of these particles makes them likely to activate at very low supersaturations, and they are present in more than sufficient number concentration to impact the microphysical structure and processes in stratocumulus clouds by serving as "giant CCN" (Feingold et al., 1999).

No significant relationship between wind speed and fine mode aerosol population type was noted. However, particles in the coarse mode range are not measured or accounted for in our cluster analysis (CCN system range: 17-500 nm), while submicron aerosol was often dominated by aerosol from other sources. Modini et al., (2015) utilized a dedicated size distribution fitting analysis that included size resolved observations of particles above 500 nm to examine primary submicron marine aerosol production. They found a primary mode with a median diameter around 200 nm and tail that extended to sizes well above 500 nm, with number concentrations of 12 +/- 2 $cm^{-3}$ during a period of low wind speeds that increased to 71 +/- 2 $cm^{-3}$ as winds increased. Concentrations differences of around 50-60 $cm^{-3}$ due to wind speed changes may not have

resulted in large enough changes to concentrations or size distributions to alter the clustering of the observed population types, which often included number concentrations that were larger by an order of magnitude or more.

## 4 Discussion

Based on this classification of the SCS remote marine boundary layer aerosol environment, a conceptual picture emerges as to the nature and sources of particles encountered during the *Vasco* 2012 cruise. A bimodal marine aerosol background was present with number concentrations usually between about 300 and 700 cm$^{-3}$ and a Hoppel minimum around 90 nm. Primary emissions via sea spray supply submicron particles consisting of a mixture of sea salt and organic components, with emitted particle diameters as small as 40 nm (Clarke et al., 2006; Keene et al., 2007; O'Dowd and Leeuw, 2007; Prather et al., 2013; Quinn et al., 2014). However, even in remote marine environments transported anthropogenic and combustion aerosol may still be an important or even dominant source of small particles (Shank et al., 2012). The background marine population identified in this study is therefore considered a background state across the remote SCS that is likely comprised of a mixture of primary marine emissions along with particles derived from anthropogenic, biomass burning, and terrestrial and marine biogenic sources throughout the region (Frossard et al., 2014). Departures from the typical range of background marine characteristics and number concentrations occurred under large influxes of aerosol from other sources, such as smoke from biomass burning regions, anthropogenic pollution from population centers or shipping, or when convection and precipitation removed much of the ambient particulate matter and created relatively clean air masses.

During the SWM when large amounts of biomass burning aerosol were being advected into the SCS, a population of aged, accumulation mode smoke particles was periodically injected into the MBL where it mixed with existing particles. When total particle concentrations were above roughly 1500 cm$^{-3}$ (Figure 3), the smoke particles dominated the background marine particles and had characteristic size distribution parameters and hygroscopicities that remained roughly constant regardless of further increases in the concentration of biomass burning particles. In situations when smoke concentrations were insufficient to dominate the background marine aerosol population, smoke mixed with the background marine population, yielding size distribution and hygroscopicity parameters in between the two types. While the background marine type was earlier noted to be impacted to some extent by background anthropogenic or terrestrial aerosol similar to impacts noted for the mixed marine type, the later was characterized by mixing with a separate, distinct aerosol population, but at levels that were insufficient to dominate the background aerosol properties.

Precipitation removal of particles that had been advected into the region or ventilation by cleaner air masses when transport pathways changed returned the environment near the surface to its background marine state. However, when extensive precipitation occurred, accumulation mode particles were removed by wet deposition to a greater extent than Aitken mode particles, leading to lower overall surface number concentrations that were dominated by smaller particles, as evidenced by the emergence of a distinct precipitation population type from the cluster analysis. Based on the two *Vasco* cruises, the cleanest periods were encountered in cold pools following the passage of squall lines with number concentrations as low as

100 to 150 cm$^{-3}$. During these periods, increased number concentrations of coarse mode aerosol were regularly observed (Figure 5; CN$_{>800}$: 5.5 ± 2.1 cm$^{-3}$), that constituted a potentially important additional source of total CCN not measured by the CCN system (0.14% SS: 44 ± 25 cm$^{-3}$; 0.38% SS: 70 ± 36 cm$^{-3}$), particularly at low supersaturations where they would be expected to activate first.

In addition to these findings, several observed phenomena during the 2012 study were similar to those from the 2011 cruise (Reid et al., 2015). In particular, rapid changes in aerosol properties and source type were noted in the wake of squall lines that left clean air masses in their wake, while longer period fluctuations on the order of days occurred as impacts from anthropogenic and smoke transport mixed with cleaner background marine and precipitation impacted air masses. As both studies were conducted in the remote marine SCS during the biomass burning season and saw similar meteorological

phenomena modulating the aerosol populations, the more detailed aerosol property results of the 2012 cruise may be representative of the general nature of changes in SCS remote marine aerosol during the SWM season. Future work in the region to compare surface properties with model results and satellite retrievals will be ultimately required to fully validate these findings.

While the cluster analysis assigned each data point to a single cluster, in reality these first four clusters could be better

described as a spectrum due to the variable impacts of mixing or meteorological processes, rather than as distinct or mutually exclusive population types. As is evident in Figure 3, overlap between these four clusters occurred in the parameter space for all nine of the measured variables used in the cluster model.

Deviations from this general picture arose when influxes of other aerosol types occurred. The additional population types each mapped out generally distinct areas in one or more of the parameters, leading to their identification by the cluster

model. That such clusters corresponded to temporally distinct periods with physical and meteorological relevance ultimately justified the use of the cluster model to classify aerosol population types and assign rough population boundaries to the parameter space.

While the spectrum of mixing between population types is relevant to the identification of impacts from various sources, additional consideration of these aerosol types against measurements in other regions is also warranted. Fresh sea spray

particles, dominated by sodium chloride (κ = 1.28), are expected to have the highest κ values, although co-emitted organic species and replacement of chlorine by uptake of acidic gases can potentially reduce hygroscopicities. Additionally, the increasing organic fractions at smaller sizes reported in sea spray aerosol (Keene et al., 2007; Prather et al., 2013; Quinn et al., 2014; Forestieri et al., 2016) lead to decreased hygroscopicities as particle size decreases. Reported hygroscopicities for aerosol in marine regions vary, with Good et al., (2010) reporting CCN-derived κ values above 1 for some background or

marine dominated MBL air masses, consistent with pure sodium chloride dominated sea salt κ values. Prather et al., (2013) generated aerosol using sea water with varying organic concentrations and associated marine biological activity, and reported CCN activation diameters at 0.2% supersaturation that correspond to κ values above 1 during periods of low organic concentration, that then dropped to κ values as low as 0.1 as organic concentration was increased. In addition, they found more organic enrichment and lower associated hygroscopicities in smaller particles. Quinn et al., (2014) explored the

relationship between organic aerosol content, particle size, and particle hygroscopicity in primary marine aerosol generated in several ocean regions. They found a similar enrichment of average aerosol organic volume fraction with decreasing particle size (40 nm dry diameter: 0.8 organic volume fraction; to 100 nm: 0.4 fraction) that corresponded with decreased hygroscopicities (40 nm: $\kappa$=0.4; 100 nm: $\kappa$=0.8), and was consistent in various ocean regions and largely independent of

biological activity as indicated by chlorophyll-a levels. Such findings are consistent with the lower Aitken mode hygroscopicities found in the background marine populations observed in this dataset, as well as the additional decreases in hygroscopicity noted in the precipitation population that had been further scrubbed of larger accumulation mode particles. Our findings of background and precipitation impacted marine aerosol $\kappa$ values generally between 0.2 and 0.6 for the Aitken mode and 0.4 and 1.0 for the accumulation mode are therefore consistent with reported hygroscopicities of background

marine aerosol that had been enriched by organic components. The presence of organics explains the noted lower hygroscopicities, as compared to what would be expected from pure sea salt aerosol, in population types that were otherwise expected to be dominated by background marine aerosol.

Aged biomass burning aerosol have often been found to have $\kappa$ values below 0.2 (Andreae and Rosenfeld, 2008; Petters et al., 2009; Engelhart et al., 2012), below the smoke population type average of approximately 0.4 in the accumulation mode

during this study. However, high concentrations of both $SO_2$ and sulfate aerosol (ammonium sulfate, $\kappa$ = 0.61) from numerous sources have been observed in the MC (Robinson et al., 2011; Reid et al., 2013). During this study, multi-day filter samples showed average sulfate concentrations between approximately 0.8 and 3 $\mu g/m^3$ at the *Vasco*, potentially increasing during periods of smoke impacts due to burning of sulfur rich peat in the region (Reid et al., 2016). The potential peat source or mixing with other sources of sulfate may explain the higher than typical $\kappa$ values observed for aged biomass

burning aerosol in the MC.

**5 Conclusion**

This study reports ship based measurements of aerosol size distributions and CCN properties conducted as part of the first extensive, in situ aerosol measurement campaign in remote marine regions of the South China Sea/East Sea during the important Southwestern Monsoon and biomass burning season. Analysis of approximately two weeks of measurements

found aerosol characteristics consistent with those from a previous pilot study in the region during the same season, indicating that descriptions of aerosol population types and the associated meteorological and transport phenomena that modulate changes and mixing between these populations may be representative of the wider remote marine SCS during the SWM season.

Eight aerosol population types were identified in the dataset that were associated with various impacts from background

marine particles, smoke, and anthropogenic sources, as well as precipitation impacts and shorter lived events linked to influxes of VOCs or ultrafine particles. Efforts to measure or model the impact of aerosol on cloud development or atmospheric optical properties often rely on proper characterization of aerosol microphysics associated with impacts from

various aerosol sources. As such, we provided population type average values and standard deviations for aerosol size distribution and hygroscopicity properties needed to model aerosol hygroscopic growth in humid environments or cloud development. Future work with this dataset will investigate the impact of the identified aerosol population types on CCN properties including supersaturation dependent CCN concentration needed to model development of different types of clouds. Reutter et al., (2009) identified specific regimes of cloud development where aerosol number concentration was important using a cloud parcel model, while Ward et al., (2010) found such results may be further complicated by aerosol size and hygroscopic properties. Inclusion of both population type average properties and the range that they vary across into such a model may help constrain when various properties of the aerosol are relevant to cloud development in the SCS. Additionally, differences in aerosol population type are expected to be relevant to studies of radiative transfer, optical propagation through the atmosphere, and satellite retrievals in sub-saturated marine environments where differences in particle number concentration, size, hygroscopicity, index of refraction, and relative humidity all affect the interaction radiation with particles in complex ways.

Lastly, while specific observed aerosol population types were identified in this dataset, additional open questions remain regarding the relative importance of various sources and transport pathways of aerosol into remote MBL air masses and their impact on aerosol populations. Since the surface-based observations provide only a portion of the observations needed to construct a true aerosol budget for the MBL, the degree to which MBL aerosol may be impacted by mixing down from a reservoir aloft was not clear. Future airborne aerosol campaigns in the region may be useful to shed light on this important topic.

**Acknowledgments**. Funding for this research cruise and analysis was provided from a number of sources. Vasco ship time procurement was provided by the NRL 6.1 Base Program via an ONR Global grant to the Manila Observatory. Core funding for this effort was from Office of Naval Research 322 under Award Number N00014-16-1-2040 and the Naval Research Enterprise Internship Program (NREIP). Funding for NRL scientist participation was provided by the NRL Base Program and ONR 35. This material is based upon research supported by the Office of Naval Research under Award Number N00014-16-1-2040, and by the Colorado State University Center for Geosciences/Atmospheric Research (CG/AR). We are most grateful to the Vasco ship management and crew, operated by Cosmix Underwater Research Ltd, (esp. Luc Heymans and Annabelle du Parc), the Manila Observatory senior management (esp. Antonia Loyzaga and Fr. Daniel McNamara), and the US State Department/ Embassy in Manila (esp. Maria Theresa Villa and Dovas Saulys). We would also like to thank the two anonymous reviewers and the Editor for their insightful comments and helpful suggestions.

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

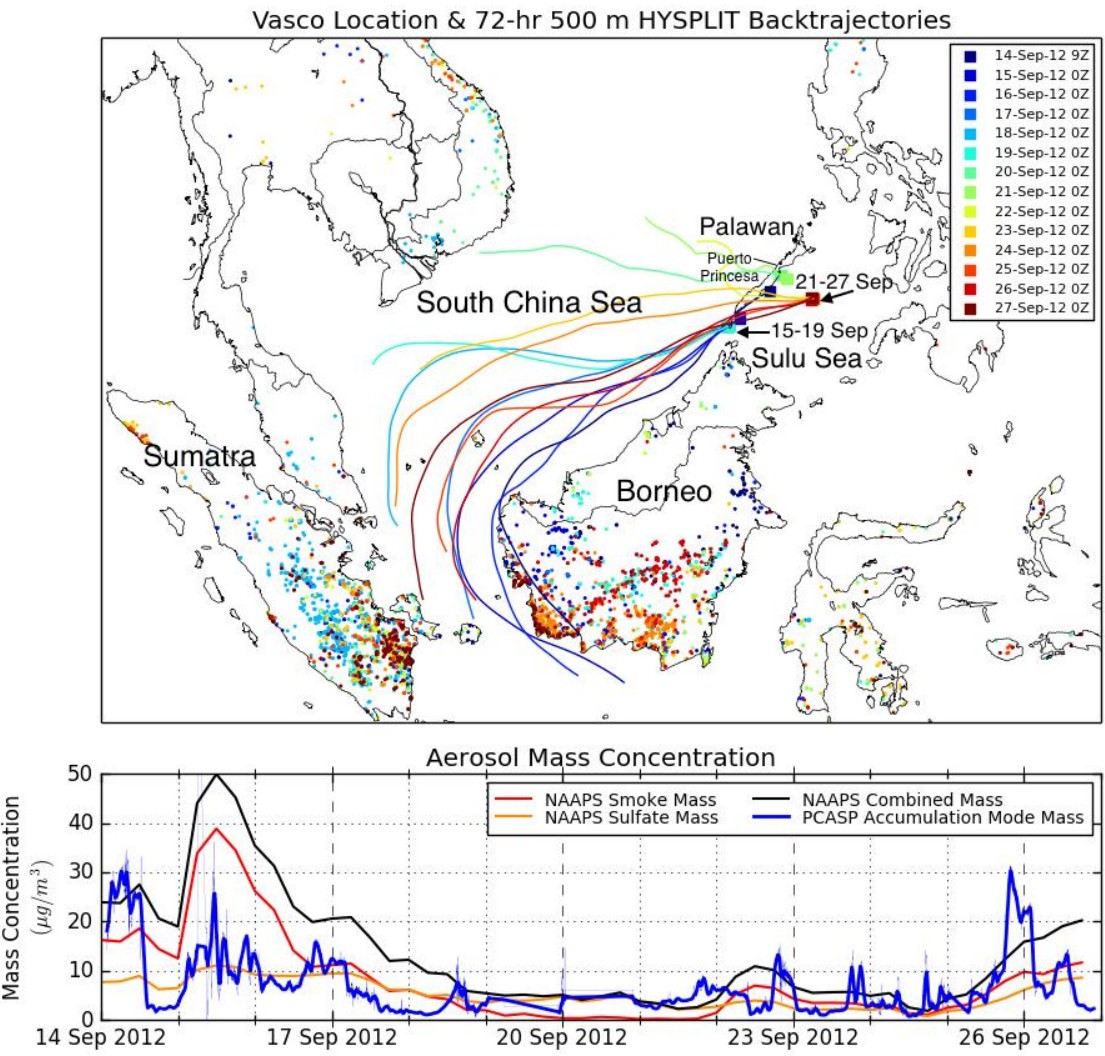

**Figure 1: (a)** *Vasco* **cruise locations (squares) and 72-hour, 500 m HYSPLIT backtrajectories; MODIS fire detections (dots) from Terra and Aqua are included for each day (color coded) during the sampling period. (b) PCASP reconstructed accumulation mode (125nm – 800nm) mass concentration (assumed density 1.4 μg m⁻³) and NAAPS estimated smoke and sulfate mass concentration along the** *Vasco* **ship-track.**

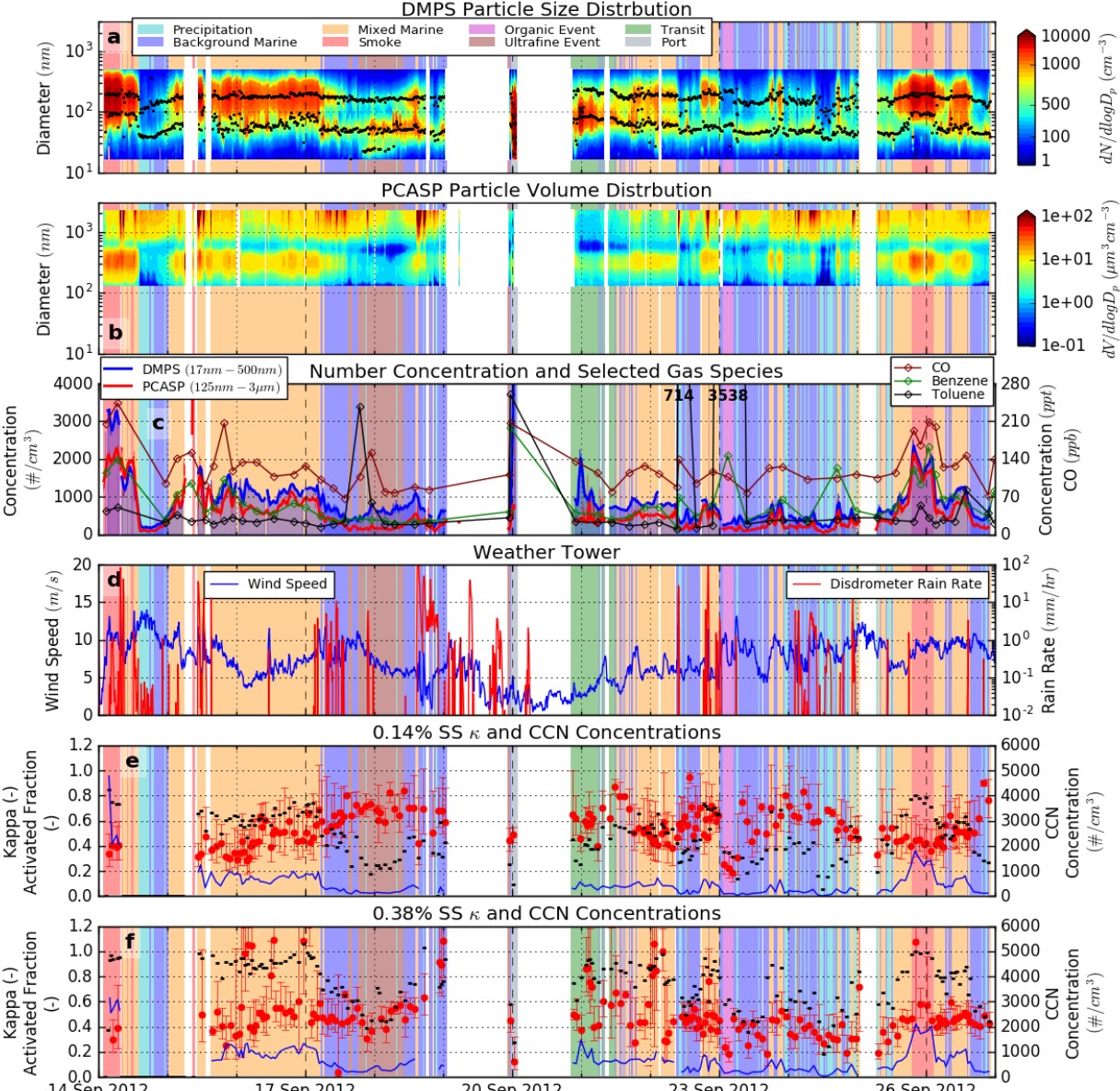

**Figure 2: Timelines of measured and derived variables during *Vasco* 2012 cruise.** In all figures, background colors correspond to aerosol type classification from the cluster analysis, as indicated in the legend in panel a. (a) $dN/dlogD_p$ spectra from the CCN system measurements with black dots at best-fit modal median diameters; (b) $dV/dlogD_p$ spectra derived from the PCASP measurements; (c) total number concentrations measured by the CCN system (blue; shaded below for contrast) and the PCASP (red), with 60 minute boxcar average smoothing; gas canister grab sample concentrations for carbon monoxide, benzene, and toluene are shown on the right axis with colored numbers indicating points above the upper scale extent; (d) wind speed and disdrometer rain rate from the *Vasco* weather tower. (e & f) κ parameter (red) and CCN concentrations (blue) for 0.14% and 0.38% supersaturation settings (corresponding approximately to accumulation and Aitken modes, respectively), with total activated particle number fractions ($CCN_{SS\%}$ / $CN_{Total}$) bars in grey. Error bars on κ data points indicate the κ values associated with 25% / 75% activated fraction curve fits.

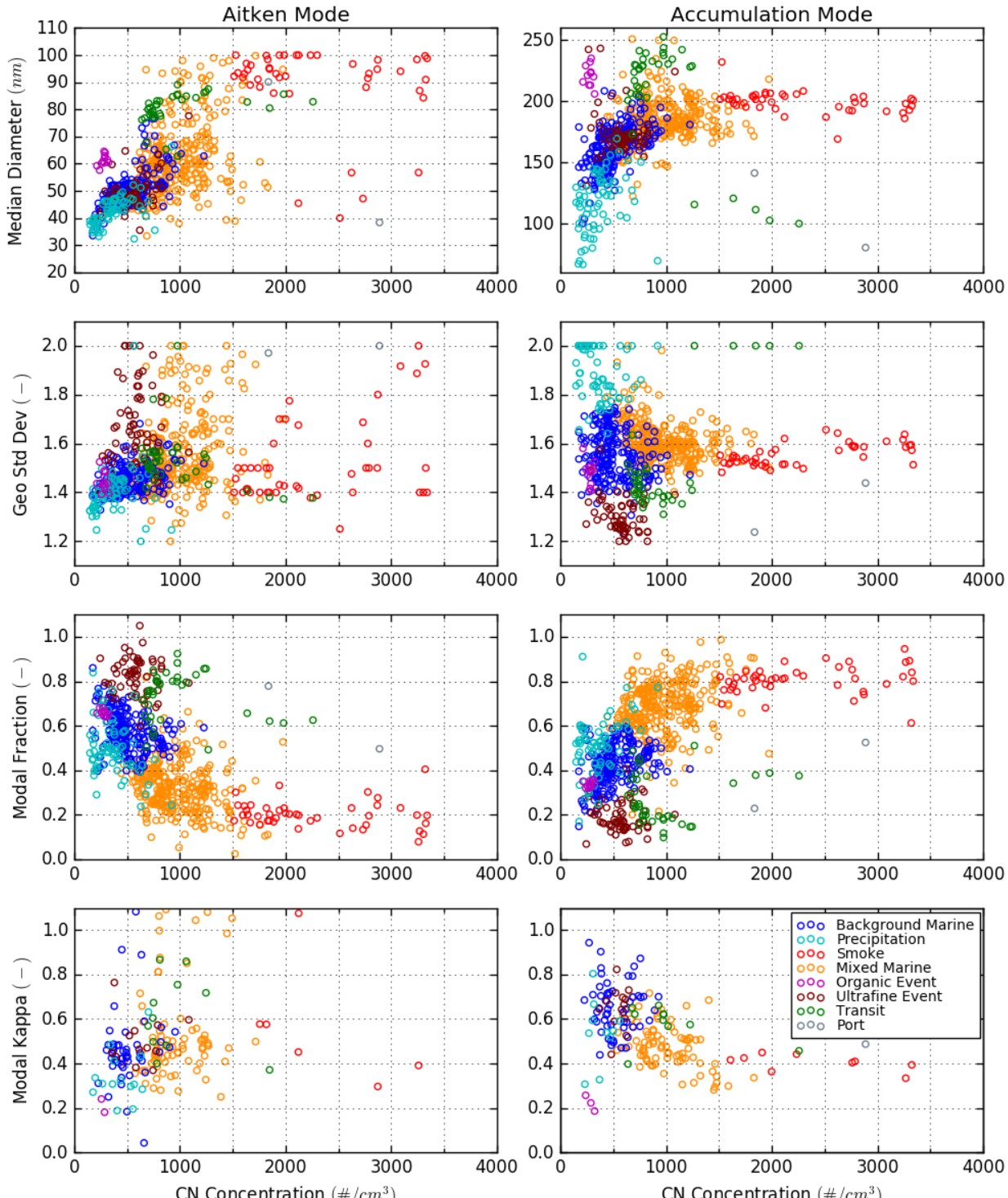

**Figure 3: Parameterized variable values for Aitken and accumulation modes (median diameter, geometric standard deviation, modal fraction) at each of the 15 minute data points during the study, along with $\kappa$ values for data points at CCNc superstation set points of 0.38% (Aitken mode) and 0.14% (accumulation mode). Each data point is colored according to the cluster type to which it was classified.**

# Normalized dN/dlogDp

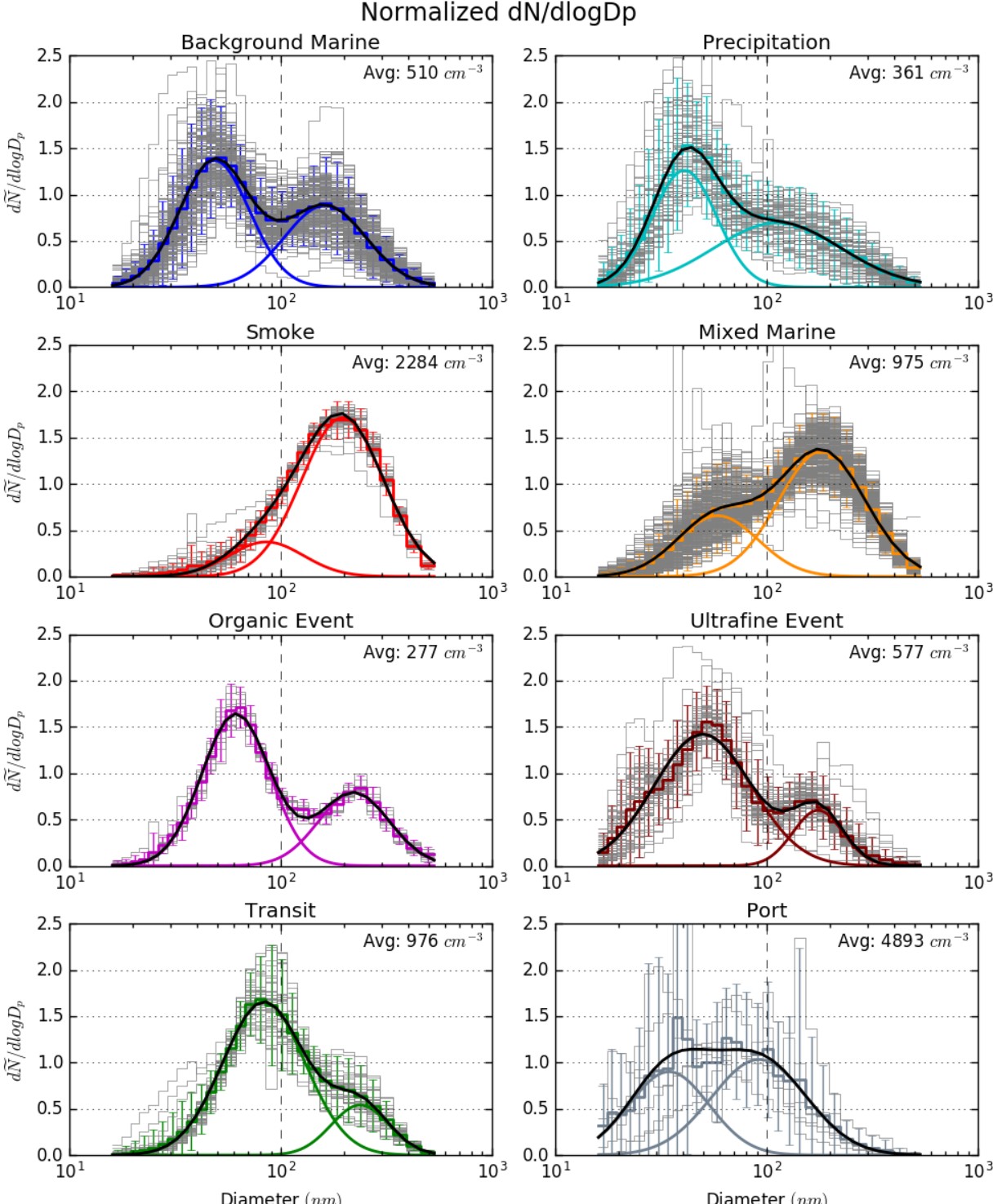

**Figure 4: Normalized dN/dlogD$_p$ particle size distributions for each spectrum within identified aerosol population types (grey), the associated average bin values with error bars at 95% confidence interval (colored step lines; 1.96 * bin standard deviation), and best fit lognormal modes (colored curves) with bimodal fit (black). The average particle number concentrations of data points within each population type are listed.**

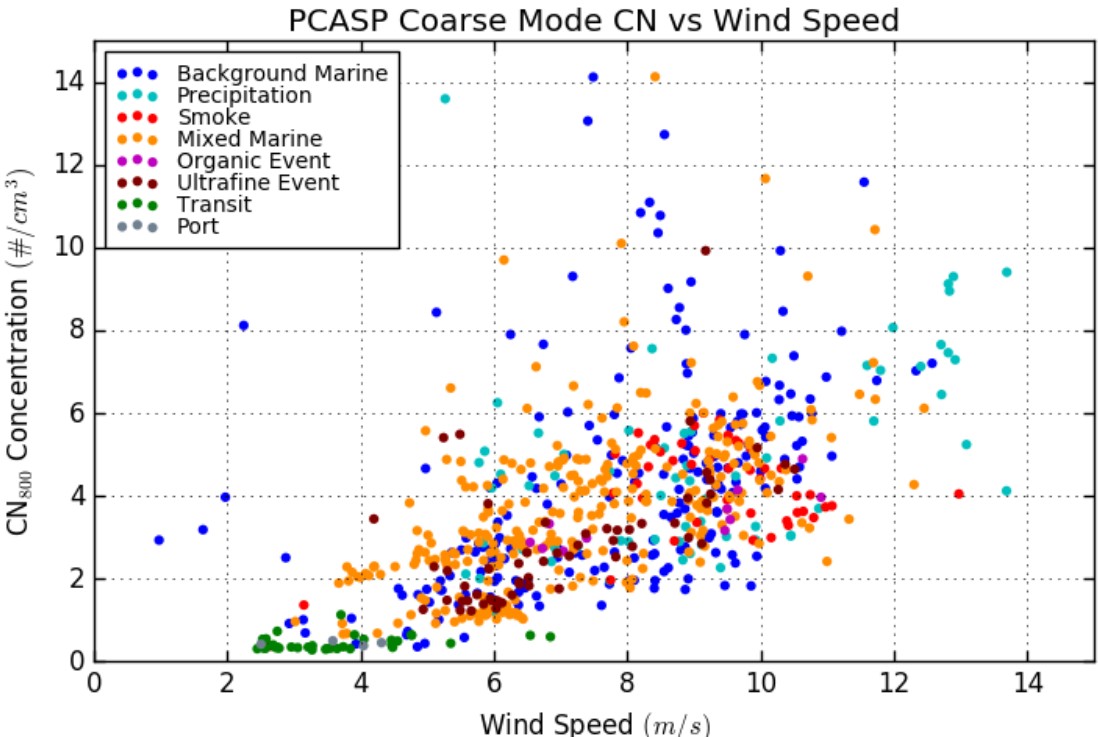

**Figure 5: Number concentrations of coarse mode particles (D$_p$ > 800 nm) measured by the PCASP as functions of local surface wind speed measured by the onboard *Vasco* weather station. Each point was averaged over the same approximately 15 minute time period as for the CCN system measurements, and is colored by the aerosol type as described in the text.**

**Table 1: Aerosol population type parameters used for clustering and the resulting average values (standard deviations in grey parentheses) for each identified population.**

| Population Type | Total Number Concentration (# cm$^{-3}$) | Aitken Mode | | | | Accumulation Mode | | | |
|---|---|---|---|---|---|---|---|---|---|
| | | Median (nm) | Geometric Std Dev | Number Fraction | Kappa | Median (nm) | Geometric Std Dev | Number Fraction | Kappa |
| 1: Back. Marine | 510 (181) | 50 (7) | 1.45 (0.05) | 0.57 (0.08) | 0.46 (0.17) | 162 (18) | 1.55 (0.10) | 0.42 (0.09) | 0.65 (0.11) |
| 2: Precipitation | 361 (164) | 42 (5) | 1.40 (0.10) | 0.50 (0.12) | 0.34 (0.11) | 115 (27) | 1.91 (0.20) | 0.51 (0.12) | 0.54 (0.14) |
| 3: Smoke | 2280 (606) | 89 (15) | 1.53 (0.17) | 0.20 (0.06) | 0.56 (0.25) | 199 (9) | 1.55 (0.04) | 0.81 (0.06) | 0.40 (0.03) |
| 4: Mixed Marine | 975 (271) | 62 (13) | 1.54 (0.18) | 0.32 (0.13) | 0.54 (0.23) | 184 (17) | 1.61 (0.08) | 0.68 (0.12) | 0.48 (0.10) |
| 5: Organic Event | 277 (30) | 61 (2) | 1.45 (0.04) | 0.66 (0.01) | 0.21 (0.03) | 221 (8) | 1.48 (0.05) | 0.34 (0.01) | 0.22 (0.03) |
| 6: Ultrafine Event | 577 (158) | 50 (6) | 1.69 (0.16) | 0.82 (0.08) | 0.50 (0.10) | 174 (19) | 1.30 (0.07) | 0.19 (0.06) | 0.65 (0.09) |
| 7: Transit | 976 (384) | 80 (6) | 1.53 (0.12) | 0.73 (0.11) | 0.62 (0.16) | 209 (42) | 1.50 (0.21) | 0.26 (0.11) | 0.58 (0.08) |
| 8: Port | 4890 (2550) | 42 (22) | 1.62 (0.37) | 0.49 (0.18) | 0.13 (-) | 87 (26) | 1.57 (0.22) | 0.53 (0.18) | 0.49 (-) |

**Table 2: Average values (standard deviations in grey parentheses) for identified aerosol population types. Shown are number of CCN system data points classified as each type, total number concentrations for the PCASP (125 nm–3 µm) and CCN system (17–500 nm), CCN number concentrations and activated fractions for each CCNc supersaturation set point, and measured κ values for the accumulation mode (0.14% SS) and Aitken mode (0.38% SS) set points.**

| Population Type | # CCN Meas. (#) | PCASP Number (#/cm$^3$) | CCN system Number (#/cm$^3$) | CCN (#/cm$^3$) | 0.14% SS Act Frac (-) | κ (-) | CCN (#/cm$^3$) | 0.38% SS Act Frac (-) | κ (-) | CCN (#/cm$^3$) | 0.53% SS Act Frac (-) | CCN (#/cm$^3$) | 0.71% SS Act Frac (-) | CCN (#/cm$^3$) | 0.85% SS Act Frac (-) |
|---|---|---|---|---|---|---|---|---|---|---|---|---|---|---|---|
| 1: Back. Marine | 214 | 231 (111) | 510 (181) | 213 (101) | 0.38 (0.09) | 0.65 (0.11) | 320 (148) | 0.60 (0.12) | 0.46 (0.17) | 416 (194) | 0.74 (0.11) | 444 (239) | 0.81 (0.09) | 480 (210) | 0.87 (0.05) |
| 2: Precipitation | 67 | 142 (79) | 361 (164) | 96 (58) | 0.24 (0.11) | 0.54 (0.14) | 243 (135) | 0.48 (0.15) | 0.34 (0.11) | 352 (175) | 0.65 (0.15) | 265 (82) | 0.71 (0.09) | 228 (100) | 0.79 (0.03) |
| 3: Smoke | 44 | 1800 (273) | 2280 (606) | 1720 (388) | 0.72 (0.04) | 0.40 (0.03) | 2340 (480) | 0.93 (0.02) | 0.56 (0.25) | 1990 (359) | 0.97 (0.02) | 2080 (396) | 0.98 (0.05) | 2150 (523) | 0.99 (0.02) |
| 4: Mixed Marine | 294 | 689 (295) | 975 (271) | 591 (201) | 0.58 (0.08) | 0.48 (0.10) | 827 (270) | 0.83 (0.07) | 0.54 (0.23) | 861 (247) | 0.89 (0.10) | 876 (244) | 0.94 (0.06) | 893 (271) | 0.96 (0.05) |
| 5: Organic Event | 11 | 151 (19) | 277 (30) | 88 (10) | 0.31 (0.02) | 0.22 (0.03) | 144 (9) | 0.53 (0.01) | 0.21 (0.03) | 182 (26) | 0.72 (0.01) | 268 (56) | 0.89 (0.14) | 257 (24) | 0.93 (0.07) |
| 6: Ultrafine Event | 59 | 163 (58) | 577 (158) | 138 (45) | 0.25 (0.06) | 0.65 (0.09) | 361 (172) | 0.56 (0.11) | 0.50 (0.10) | 373 (168) | 0.65 (0.12) | 439 (163) | 0.72 (0.07) | 473 (147) | 0.79 (0.10) |
| 7: Transit | 36 | 311 (44) | 976 (384) | 363 (87) | 0.37 (0.06) | 0.58 (0.08) | 772 (263) | 0.81 (0.09) | 0.62 (0.16) | 832 (423) | 0.87 (0.05) | 877 (370) | 0.90 (0.02) | 878 (195) | 0.95 (0.03) |
| 8: Port | 6 | 671 (210) | 4890 (2550) | 251 (-)* | 0.09 (-)* | 0.49 (-)* | 1126 (-)* | 0.26 (-)* | 0.13 (-)* | 3936 (-)* | 0.40 (-)* | 1742 (289) | 0.57 (0.22) | 2080 (-)* | 0.45 (-)* |
| **All Types** | **731** | **503 (455)** | **851 (677)** | **450 (388)** | **0.47 (0.16)** | **0.54 (0.14)** | **675 (516)** | **0.72 (0.17)** | **0.50 (0.21)** | **698 (555)** | **0.79 (0.15)** | **724 (512)** | **0.85 (0.13)** | **723 (502)** | **0.90 (0.10)** |

* Only one datapoint; Note that Port measurements fluctuated as the Vasco entered port