# Peer review of "Size-resolved aerosol and cloud condensation nuclei (CCN) properties in the remote marine South China Sea, Part 1: Observations and source classification"

_Atmospheric Chemistry and Physics, 2016_

## Referee Comment (RC1) · Anonymous Referee #1 · 19 Aug 2016

Overall.

This paper is clear and well written. Measurements were made in the remote South China Sea/East Sea during the previously unmeasured Southwestern Monsoon and biomass burning season, making the measurements unique. The analysis is described in detail, and the authors took care to provide all information necessary for a clear picture of the project. The classification of the aerosol types is interesting and relevant for this audience. However, the discussion or conclusion could be expanded to include the implications of these results.

[Figure]

General Comments.

More discussion of the back trajectories would be useful. Why was 72-hours back chosen? Was more than one back trajectory per day run? How did the patterns change based on time of day? What time were the daily back trajectories calculated at, or are they representative of a daily average? The height of 500 m is discussed – how do other heights compare?

For the cluster identification in Section 3.3, more details could be added to each group to further explain the identification. While some of the events used to identify the clusters were discussed in an earlier section, they could be added here for emphasis and clarity.

Correlations with wind speed and other indicators (salt concentrations, etc.) could be useful in identifying the Background Marine cluster. It is discussed later that the coarse mode concentrations increase with increasing wind speed. Was this only for the coarse mode particles? The Kappa value for background marine is also much lower than that of salt. How does that compare to other marine Kappa values, especially those with high O/C or hydrophilic compounds in the organic fraction?

More discussion of the impact and implications of these results should be included. These clusters of aerosol types are identified, but what does that mean for other aerosol or cloud properties in the area or even globally?

The Conclusions section is somewhat short and vague. A couple more sentences with specific conclusions would be useful to summarize the main points. Include some implications of the results.

Specific Comments.

Page 3, Line 8: How was sampling shut down? Add a reference to Reid et al., if it is described there.

Page 3, Line 20: Define "DMT PCASP X2"

[Figure]

Page 4, Line 12: Did the properties of the aerosol remain constant in that 2 hour time period?

Page 5, Line 23: Include the parameters input into the clustering here. Did any of the variables dominate the clustering (i.e. distribution variables vs. Kappa)?

Page 8, Line 2: Was there anything besides the bimodal distribution that indicated this was background marine air?

Page 9, Line 14: What was the source of the organic event?

Page 12, Line 14: If the background marine is comprised of primary marine and anthropogenic/biogenic sources, how is that type different from the Mixed Marine cluster?

Figure 1: There appear to be more square markers in the legend than in the figure. Are the square markers representative of the ship location on only the days matching the legend, or is it a range of days? It seems like it is a range of days. If that is the case, a different marker system or extended legend would be useful (i.e. circle for first 3 days, square for others, etc.). Some location labels would also be helpful for the discussion (i.e. Borneo, Sumatra, etc.).

Figure 2: The background colors in the panels should all be the same (a and b are darker than the others). It is hard to associate the variability of the Kappa parameter with the different aerosol types in e and f.

Table 1: The table is really small and should rotated or condensed for publication.

---

## Referee Comment (RC2) · Anonymous Referee #2 · 30 Aug 2016

General comments:

The paper presents a comprehensive and state-of-the-art aerosol size distribution and CCN dataset taken during a research cruise in the South China Sea. The aerosol cluster analysis in combination with the attribution of aerosol types to air masses and sources is interesting and relevant. The report of representative hygroscopicity values is very useful for the community. The paper is written in a good and clear language, but could be made more concise by dropping repetitions of observations. The scientific content is largely descriptive and should be made stronger by putting the results into

context. This is particularly important for the following sections:

Conclusions: The meaning of the results should be carved out better and the conclusions made more specific. E.g. p.13, line 31: How exactly do the authors arrive at the conclusions that results are regionally and temporally representative (p. 13, line 31), and what are the further implications? What time period is meant by "temporally" – the season, a whole year? What "previous study"?

Aerosol Hygroscopicity: It would be nice if more of the CCN data and their significance were discussed and if they were put into more context. What supersaturations are expected to be relevant in the regionally typical cloud cover; are the ones chosen here representative? It would be nice if the impact of biomass burning and anthropogenic emissions on all CCN parameters (not just kappa) were carved out a little more.

Introduction: The first paragraph (especially up to line 11) should be made more specific. For example, lines 7 and 8: Which additional questions? Representativeness of which results? Also, the first sentence of the introduction is somewhat unfortunate. It begs the question: Why is it important to assess aerosol properties there in the first place? The region and its significance to aerosol research need to be more clearly introduced to the reader.

Discussion: Currently, the section contains mostly repetitions of observations discussed earlier in the paper, and a couple new observations and interpretations. It should be re-worked such that it ties up the results in a way that leads to the conclusions. Alternatively, it could simply be eliminated (the new observations could be discussed in the "results" section).

Specific comments:

Abstract, line 19: this needs to be re-worded. Right now, the reader might get the impression that the "additional onboard (. . .) model products for the region" somehow entered the cluster analysis.

[Figure]

Page 3, lines 16-19 should be moved to section 2.2. What time of day was chosen for the HYSPLIT trajectory arrival? How were the other initial conditions (especially arrival height) motivated?

Page 3, line 21: What was the height of the mast above the water?

Page 3, line 29: What % relative humidity was ensured with this system?

Page 4, line 4 and 8: When and where were the calibrations performed?

Page 4, line 10: Can you comment on the stability of the supersaturation settings throughout the cruise?

Page 4, line 25: What was the temporal resolution of the filter samples? Are the analysis methods described in more detail in Reid et al., 2016? If so, the reference should be added to this sentence, too.

Page 5: please state the specific products used in this study (for example, where does the AOD mentioned in page 6, line 13, come from)?

Page 6, line 8: How were "surface winds" averaged, over what period or area?

Page 6, line 16: Why was only this one density used - fire emissions were presumably not the only coarse aerosol type in the region?

Page 6, line 28-29: It would be helpful to label these locations in the map in Figure 1.

Page 7, line 3: perhaps specify "biomass burning smoke". Are there any filter analyses for this period?

Page 9, line 7: it would be nice to see the tri-modal fit included in Figure 4

Page 10, lines 23 – 33: This paragraph should be moved into the introduction. If a comparison of literature kappa values to this study's was intended, this comparison should be done more directly, rather than expecting the reader to jump back and forth between paragraphs.

Page 11, lines 23: -29: How can it be stated that the Aitken mode during the biomass burning period be derived from the background aerosol, when it is actually more hygroscopic than the Aitken mode of the marine background and precipitation clusters? Why would the biomass influence be confined to particles >100nm? (Surely not all biomass burning particles are primary?)

Language/typos:

Abstract, line 20: "aerosol population that" should be "aerosol population and"

Page 2, line 31: "in situ" is not the right expression here, I believe. "In the area" or "in the SCS" would be better.

Page 4, line 15 and 16: choose one, "CCN activation spectrum" or "activated fraction spectrum"

Page 4, line 25: "that were analyzed" should be "and were analyzed".

Page 4, line 14: A new sentence would be better than the hyphenation-clause combination: "number concentration ). Two modes were identified as the best fit..."

Page 4, line 16 and line 25: Start a new sentence instead of the hyphenation.

Page 4, line 22: "shortest" instead of "closest"?

Page 8, line 31: Start a new sentence after "(Figure 2d)".

Page 11, line 12: "lower (....) than in the precipitation (...) populations"

Page 12, line 1: What does "from the entire study" mean?

Caption of figure 4: "each spectrum"

Figures/Tables:

Figure 1: The tick marks on the time axis would be easier to identify if they pointed outward

Table 1: The font size is too small.

[Figure]

---

## Author Comment (AC1) · 29 Oct 2016

We would like to thank the reviewer for their thorough review and helpful suggestions. Each of the comments is listed below in bold text, with our response and corrections following each of them in plain text. We have made a number of small corrections to the manuscript for each of the specific comments as noted. In addition, we have conducted a general reorganization of the results and discussion sections pursuant to the general comments. We have included relevant updated sections in response to general comments, however it may be more effective to reference the marked-up manuscript included at the end of this document to fully address the noted concerns.

**Overall.**
**This paper is clear and well written. Measurements were made in the remote South China Sea/East Sea during the previously unmeasured Southwestern Monsoon and biomass burning season, making the measurements unique. The analysis is described in detail, and the authors took care to provide all information necessary for a clear picture of the project. The classification of the aerosol types is interesting and relevant for this audience. However, the discussion or conclusion could be expanded to include the implications of these results.**

**General Comments.**
**More discussion of the back trajectories would be useful. Why was 72-hours back chosen? Was more than one back trajectory per day run? How did the patterns change based on time of day? What time were the daily back trajectories calculated at, or are they representative of a daily average? The height of 500 m is discussed – how do other heights compare?**
We have moved the description of use of the HYSPLIT model to the end of section 2.2 and amended the paragraph to address these questions as follows:
"The NOAA Hybrid Single Particle Lagrangian Integrated Trajectory (HYSPLIT) Version 4.9 model (Draxler et al., 1999; Draxler and Hess, 1997, 1998) was used to generate daily 72-hour backtrajectories (spawned at 0Z, 8 AM local) from the *Vasco* location with arrival heights of 500m to indicate likely marine boundary layer transport patterns. The GDAS1, 1° x 1° HYSPLIT meteorological dataset was used to drive the model. Trajectory paths were found to be largely influenced by synoptic scale changes in the regional meteorological state of the atmosphere, with no substantial differences due to arrival time of day. Arrival heights between 100 m and 3000 m were examined. Trajectories with arrival heights below 1000 m were generally consistent and representative of boundary layer transport (Atwood et al., 2013; Xian et al., 2013), while higher heights tended to be increasingly influenced by free troposphere transport pathways with a more westerly component. As such, 500 m was selected to be representative of general shifts in synoptic scale boundary layer transport pathways, though more complex vertical interactions and mixing from aloft are a potential influence in the region (Atwood et al., 2013). Trajectory lengths of 72 hours were found to be sufficient to demonstrate general transport path differences between ocean dominated regions of the central portion of the SCS and more terrestrially influenced regions that passed closer to Borneo and Sumatra."

**For the cluster identification in Section 3.3, more details could be added to each group to further explain the identification. While some of the events used to identify the clusters were discussed in an earlier section, they could be added here for emphasis and clarity.**

In order to better organize the results in line with several comments, we removed Section 3.2 "Analysis of Daily Observations" and have combined all of the results into the new section 3.2 with relevant descriptions of all results important to identification and description of each cluster. The new section 3.2 is included here, while a marked-up version with changes noted can be found in the full marked-up manuscript at the end of this document:

[revised manuscript text omitted]

**Correlations with wind speed and other indicators (salt concentrations, etc.) could be useful in identifying the Background Marine cluster. It is discussed later that the coarse mode concentrations increase with increasing wind speed. Was this only for the coarse mode particles?**
Wind speed was not found to be correlated to accumulation or Aitken mode particle properties in this analysis. Rather, the noted relationship of wind speed with coarse mode particles (largely thought to be sea spray aerosol, consistent with the finding of similar studies, e.g. O'Dowd and Leeuw, (2007)) was in fact independent of the fine mode particles or population type as can be generally seen in Figure 5.
This has been clarified in the paragraph at the end of the new section 3.2 as follows:
"Finally, throughout the study coarse mode particles with diameters larger than about 800 nm were consistently observed in the PCASP volume distributions (Figure 2b). Concentrations of particles in this size range increased with increasing wind speed (Figure 5), consistent with generation of sea spray aerosol due to bubble breaking and wave action (O'Dowd and Leeuw, 2007). In addition, no significant relationship between wind speed and fine mode aerosol population type was noted."

**The Kappa value for background marine is also much lower than that of salt. How does that compare to other marine Kappa values, especially those with high O/C or hydrophilic compounds in the organic fraction?**
We have updated a paragraph in the discussion section to discuss both the hygroscopicity of pure sea salt, as well as enhanced organic fractions that have been reported as size decreases in sea spray aerosol, and implications for our population types:
"Fresh sea spray particles, dominated by sodium chloride ($\kappa$ = 1.28), are expected to have the highest $\kappa$ values (Good et al., 2010), although co-emitted organic species and replacement of chlorine by uptake of acidic gases can potentially reduce hygroscopicities. Additionally, increasing organic fractions at smaller sizes have been reported in sea spray aerosol (Prather et al., 2013), leading to decreased hygroscopicities as organic fraction increases. Such findings are

consistent with the lower Aitken mode hygroscopicities found in the background marine populations observed in this dataset, as well as the additional decreases in hygroscopicity noted in the precipitation population that had been further scrubbed of larger accumulation mode particles."

**More discussion of the impact and implications of these results should be included. These clusters of aerosol types are identified, but what does that mean for other aerosol or cloud properties in the area or even globally?**

**The Conclusions section is somewhat short and vague. A couple more sentences with specific conclusions would be useful to summarize the main points. Include some implications of the results.**

We have added more on the implications of the identified aerosol population types, specifically in regards to better understanding cloud development and radiative transfer in the remote marine SCS region. We have amended the second paragraph in the conclusion to read:

"Eight aerosol population types were identified in the dataset that were associated with various impacts from background marine particles, smoke, and anthropogenic sources, as well as precipitation impacts and shorter lived events linked to influxes of VOCs or ultrafine particles. Efforts to measure or model the impact of aerosol on cloud development or atmospheric optical properties often rely on proper characterization of aerosol microphysics associated with impacts from various aerosol sources. As such, we provided population type average values and standard deviations for aerosol size distribution and hygroscopicity properties needed to model aerosol hygroscopic growth in humid environments or cloud development. Future work with this dataset will investigate the impact of the identified aerosol population types on CCN properties including supersaturation dependent CCN concentration needed to model development of different types of clouds. Reutter et al., (2009) identified specific regimes of cloud development where aerosol number concentration was important using a cloud parcel model, while Ward et al., (2010) found such results may be further complicated by aerosol size and hygroscopic properties. Inclusion of both population type average properties and the range that they vary across into such a model may help constrain when various properties of the aerosol are relevant to cloud development in the SCS. Additionally, differences in aerosol population type are expected to be relevant to studies of radiative transfer, optical propagation through the atmosphere, and satellite retrievals in sub-saturated marine environments where differences in particle number concentration, size, hygroscopicity, index of refraction, and relative humidity all affect the interaction radiation with particles in complex ways."

**Specific Comments.**

**Page 3, Line 8: How was sampling shut down? Add a reference to Reid et al., if it is described there.**

The ship sampling ports were installed near the front of the ship such that when relative wind was from across the bow, sampled air was clear of self-sampling. During long periods when winds were not across the bow instrumentation was shut down. In short duration cases of incompatible winds, the times were invalidated and removed from the dataset. The text has been updated to reflect this, and further descriptions of these methods included in Reid et al., (2016) have now been cited. The sentence now reads:

"Sampling occurred throughout the cruise, but aerosol measurements were shut down or invalidated and removed from the dataset during periods when representative sampling could not be achieved (i.e., measured relative wind not from over the ship bow, leading to potential self-sampling; see Reid et al., (2016) for more details)."

**Page 3, Line 20: Define "DMT PCASP X2"**
This has been corrected to "Passive Cavity Aerosol Spectrometer Probe (PCASP)"

**Page 4, Line 12: Did the properties of the aerosol remain constant in that 2 hour time period?**
Conditions in the SCS were observed to occasionally change rapidly, including during the passage of gust fronts that could result in changes on the order of minutes (Reid et al., 2016). In these cases, the aerosol would in fact not be constant over a full 2 hour scan. However, the nature of the cluster model classification allows for each data point to be handled independently from others that occurred during this nominal 2 hour scan—the caveat being that while each 15 minute data period includes a full size scan, CCN measurements at only one supersaturation are included. In some cases, the aerosol would change on a faster time scale than a single 15 minute scan, however, such situations were rare and would effectively add to noise in the cluster results.

As to the wider question of how a change in aerosol across a 2 hour scan impacts the results, we have added clarification in two locations. First, at the end of the paragraph noted in this comment, we have added some additional explanation on how the 15 minute scans were conducted. Namely, the two lowest supersaturations were run more often than the three higher ones, as more reliable measurements of kappa could be obtained for the relatively larger marine aerosols. As such, the data were better interpreted as 15 minute data points with each point missing some CCN data, rather than a full two hour scan. The added sentence is:
"In addition, rather than continuously running a full two-hour scan across all supersaturation settings, individual scans (approximately 15 minutes) were run more often for the 0.14% and 0.38% settings to take advantage of this outcome." ["this outcome" referring to valid hygroscopicity measurements at these two supersaturations noted earlier in the paragraph.]

Second, we added some additional explanation to the clustering methodology in the second paragraph of section 2.3 in order to clarify that modal hygroscopicity data was included for data points for which it was available, and treated as missing for data points when it was not.
"As hygroscopicity data was available for at most one mode (during the 0.14% or 0.38% supersaturation scans), Aitken and accumulation mode hygroscopicity was treated as missing for data points without this information. In order to account for missing data and adjust all clustering variables to the same scale, each variable was first standardized to a mean of zero and standard deviation of one, with missing data points imputed to a value a zero (the mean value). As a result, the clustering distance function was insensitive to missing data, but still included information on hygroscopicity when available."

**Page 5, Line 23: Include the parameters input into the clustering here. Did any of the variables dominate the clustering (i.e. distribution variables vs. Kappa)?**

We have included a table of clustering variables used as input parameters to the cluster model, along with each cluster's mean and intra-cluster standard deviation to provide some of this information as the new Table 1. Determination of the relative importance of input parameters in a cluster analysis is subject to a number of uncertainties that make a quantitative determination of this value difficult. While analysis of variance methods may allow for comparison of the variance of cluster means against intra-cluster variance using an F-test, such results rely on the assumption of normality of each of these types of variance and produce results based on different sets of data for each variable. There is no guarantee that the cluster means generated by a cluster analysis are normally distributed about the variable mean, and the sometimes small number of cluster members makes it difficult to reject the null hypothesis that cluster values are normally distributed. As a result, we do not include an F-test metric that might provide some indication of relative importance of the variables to the cluster analysis in this table.

Similarly, conducting additional cluster analyses after having removed of one of the input variables may provide some indications of relative importance, but evaluation of the similarity of the results to the original clusters suffers from the lack of a specific underlying model that describes the results (i.e. we cannot quantitatively evaluate the difference between two cluster results). As such, quantitative determination of the relative importance of each variable in the clustering is not feasible and we therefore did not speculate as the relative importance of each parameter. Instead we relied on ensuring parameters were relevant to the analysis, and verified that the results maintained a physically realistic interpretation consistent with other measurements.

The new Table 1 is:

Table 1: Aerosol population type parameters used for clustering and the resulting average values (standard deviations in grey parentheses) for each identified population.

| Population Type | Total Number Concentration (# cm$^{-3}$) | Aitken Mode | | | | Accumulation Mode | | | |
|---|---|---|---|---|---|---|---|---|---|
| | | Median (nm) | Geometric Std Dev | Number Fraction | Kappa | Median (nm) | Geometric Std Dev | Number Fraction | Kappa |
| 1: Back. Marine | 510 (181) | 50 (7) | 1.45 (0.05) | 0.57 (0.08) | 0.46 (0.17) | 162 (18) | 1.55 (0.10) | 0.42 (0.09) | 0.65 (0.11) |
| 2: Precipitation | 361 (164) | 42 (5) | 1.40 (0.10) | 0.50 (0.12) | 0.34 (0.11) | 115 (27) | 1.91 (0.20) | 0.51 (0.12) | 0.54 (0.14) |
| 3: Smoke | 2280 (606) | 89 (15) | 1.53 (0.17) | 0.20 (0.06) | 0.56 (0.25) | 199 (9) | 1.55 (0.04) | 0.81 (0.06) | 0.40 (0.03) |
| 4: Mixed Marine | 975 (271) | 62 (13) | 1.54 (0.18) | 0.32 (0.13) | 0.54 (0.23) | 184 (17) | 1.61 (0.08) | 0.68 (0.12) | 0.48 (0.10) |
| 5: Organic Event | 277 (30) | 61 (2) | 1.45 (0.04) | 0.66 (0.01) | 0.21 (0.03) | 221 (8) | 1.48 (0.05) | 0.34 (0.01) | 0.22 (0.03) |
| 6: Ultrafine Event | 577 (158) | 50 (6) | 1.69 (0.16) | 0.82 (0.08) | 0.50 (0.10) | 174 (19) | 1.30 (0.07) | 0.19 (0.06) | 0.65 (0.09) |
| 7: Transit | 976 (384) | 80 (6) | 1.53 (0.12) | 0.73 (0.11) | 0.62 (0.16) | 209 (42) | 1.50 (0.21) | 0.26 (0.11) | 0.58 (0.08) |
| 8: Port | 4890 (2550) | 42 (22) | 1.62 (0.37) | 0.49 (0.18) | 0.13 (-) | 87 (26) | 1.57 (0.22) | 0.53 (0.18) | 0.49 (-) |

**Page 8, Line 2: Was there anything besides the bimodal distribution that indicated this was background marine air?**

We have addressed this as part of the new results section 3.2 in the background marine type description:

"1.      Background Marine: Data points associated with this cluster occurred throughout the study, typically following rain in the vicinity of the Vasco or transport from areas further removed from terrestrial regions. In addition, this type was observed following shortly after periods associated with each of the other identified clusters, often appearing as a transition between other types (Figure 2). The measured properties of this population type were similar

to the background marine aerosol reported in many prior studies (Hoppel et al., 1994; O'Dowd et al., 1997; Spracklen et al., 2007; Good et al., 2010). The population featured a bimodal size distribution with a Hoppel minimum near 90 nm, thought to be due to cloud or fog processing of marine aerosol (Hoppel et al., 1986). The inner quartile range (IQR: middle 50% of observations between the 25%-75% percentiles) of number concentrations ranged from 382 to 623 cm$^{-3}$, with on average 42% of the total number concentration residing in the accumulation mode as specified by the bimodal fit. Modal hygroscopicities were found to average 0.65 for the accumulation mode and 0.46 for the smaller Aitken mode, while activated fractions were generally moderate across the range of measured supersaturations as compared to other identified population types (Table 1). Each of these findings further reinforced the classification of this cluster as a typical background marine aerosol."

**Page 9, Line 14: What was the source of the organic event?**
We do not have enough information to determine the source of this event with certainty, however, biogenic SOA formation has been reported in marine regions of the SCS (Robinson et al., 2012). Additional detailed analysis of the gas canister and other sources of data from the cruise forthcoming in other manuscripts may yield more information on this event. We have updated the description of this type in the new section 3.2 to read:
"5.     Organic Event: An approximately four-hour period starting at 1Z on 23 Sept. had measured particle concentrations between 200 and 325 cm$^{-3}$, but with significantly (p<0.001) larger median diameters than either the precipitation or background marine types (Figure 3). Both Aitken and accumulation mode particles had among the lowest hygroscopicities measured during the cruise, with $\kappa$ values around 0.2. During this event measured concentrations of numerous VOCs were much higher than in gas canisters collected approximately 6 hours before and after it, with no associated increase in carbon monoxide (Reid et al., 2016; Figure 2c). The particles had lower hygroscopicities and larger sizes than the background marine particles observed just before this event. While the source of this event is uncertain, Robinson et al. (2012) found occasional organic aerosol above the boundary layer they attributed to biogenic Secondary Organic Aerosol (SOA) formation during an airborne campaign in the outflow regions of Borneo, while Irwin et al. (2011) reported $\kappa$ values between 0.05 and 0.37 in a terrestrial, biogenically dominated MC environment. Such a source would be consistent with the observed population, perhaps due to growth of a background marine population by condensation of organics, although we lack the ancillary data needed to establish this."

**Page 12, Line 14: If the background marine is comprised of primary marine and anthropogenic/biogenic sources, how is that type different from the Mixed Marine cluster?**
We agree that this is an important distinction requiring further explanation. As noted by Shank et al., (2014), a typical background marine aerosol, even in a remote location, still contains some amount of terrestrial or anthropogenic particles. Despite this, it is still useful to have a general description of a "typical" background marine aerosol state. Conceptually, we consider the Mixed Marine type distinct from this background type, occurring when an influx of additional particles from a specific source or plume happens—an impact that is nevertheless insufficient to dominate the background aerosol (i.e. a period when only the properties of a distinct source type are observed). Practically however, we agree that there is not a clear line of

demarcation between a background and a mixed marine aerosol. Rather, it is a spectrum wherein the amount of mixing with a separate source or plume places the population somewhere between the background marine properties and the properties of the mixing population (a feature evident in Figure 3 where the Mixed Marine type generally both overlaps and falls between the smoke and background marine types). In addition, we note at the end of the description of the Mixed Marine population in the new section 3.2 that "In addition, air masses influenced by anthropogenic pollution may have been included in this cluster as well, but without sufficiently different impacts on aerosol parameters to justify a distinct cluster." We have added additional clarification on this point to the discussion section 4 with the following sentence:

"While the background marine type was earlier noted to be impacted to some extent by background anthropogenic or terrestrial aerosol similar to impacts noted for the mixed marine type, the later was characterized by mixing with a separate, distinct aerosol population, but at levels that were insufficient to dominate the background aerosol properties."

Further, the discussion section now includes the following paragraph with further treatment of the issue of mixing between the main population types:

"While the cluster analysis assigned each data point to a single cluster, in reality these first four clusters could be better described as a spectrum due to the variable impacts of mixing or meteorological processes, rather than as distinct or mutually exclusive population types. As is evident in Figure 3, overlap between these four clusters occurred in the parameter space for all nine of the measured variables used in the cluster model."

**Figure 1: There appear to be more square markers in the legend than in the figure. Are the square markers representative of the ship location on only the days matching the legend, or is it a range of days? It seems like it is a range of days. If that is the case, a different marker system or extended legend would be useful (i.e. circle for first 3 days, square for others, etc.). Some location labels would also be helpful for the discussion (i.e. Borneo, Sumatra, etc.).**
The squares on the figure are indeed indicative of a range of days while at a stationary anchorage. We have included labels to indicate the range of days at the two anchorages, as well as additional labels of the location of Borneo, Sumatra, Palawan, the South China Sea, and the Sulu Sea.

[Figure]

**Figure 2: The background colors in the panels should all be the same (a and b are darker than the others). It is hard to associate the variability of the Kappa parameter with the different aerosol types in e and f.**
We have corrected the background colors to all have the same colors in all plots, and fixed the Kappa parameter plots to improve contrast. The grey activated fraction bars in the Kappa plots have been replaced with black markers that no longer obstruct the background colors.

[Figure]

**Table 1: The table is really small and should rotated or condensed for publication.**
We have included a rotated table in landscape orientation on the page, and will work with the publisher to ensure the new tables are better formatted for publication. If needed, we will split the table into several lines.

[revised manuscript text omitted]

**3.2 Analysis of Daily Observations**

At the start of the measurement period on 14 Sept., the *Vasco* was in transit south from Puerto Princessa, Philippines, in the middle of the island towards Balabac Island at the southern edge of Palawan aArchipelago. HYSPLIT 500 m backtrajectories on this day identified MBL transport from the southwest over burning regions in Borneo (Figure 1a). As shown in Figure 2, at the start of the campaign, high number concentrations of accumulation mode particles (1500 to 4000 cm$^{-3}$) with generally mono-modal size distributions were observed (best fit lognormal median diameters around 200 nm), consistent with expectations for aged smoke. Elevated concentrations of potassium, carbon monoxide, and benzene during this period (Reid et al., 2016, and Figure 2c) all reinforce the classification of this aerosol as smoke (Yokelson et al., 2008; Akagi et al., 2011; Reid et al., 2015). This initial smoke impacted period was followed by a drop in particle number concentrations after 12Z on 14 Sept. as a squall line passed over the *Vasco* and left a clean air mass scrubbed of many of the particles. Total particle number concentrations were among the lowest values observed during the cruise, with fewer than 200 cm$^{-3}$ measured in the 17-500 nm range. Particle number concentrations did not recover for approximately 10 hours, at which point the aerosol size spectra and concentrations resembled those measured in the air mass before the gust front passage.
The *Vasco* remained at the same anchorage at Balabac Island on the southern tip of the Palawan Archipelago until 19 Sept. Beginning at about 5Z on 17 Sept., particle number concentrations dropped, but not as low nor as rapidly as during the post-squall line event. A distinct bimodal size distribution was observed that coincided with the onset of precipitation observed at the boat (Figure 2d). Satellite visible and IR imagery (not shown) indicated that no long-lived squall line was associated with this 17 Sept. precipitation event. Beginning around 18Z on 17 Sept. number concentrations began to increase and a third

mode of particles with diameters between 20 and 30 nm was observed (Figure 2a). This mode slowly mixed into the Aitken mode as more typical bimodal distributions resumed by the middle of 18 Sept. A filter during this period showed very low potassium concentrations, with benzene among the lowest values measured during the study, indicating that biomass burning was not the likely source for this event. In addition, at the start of this period several volatile organic compound species (VOCs) were elevated as compared to periods before and after the event. Anthropogenic, shipping, and marine and terrestrial biogenic emissions are known sources of such compounds; isoprene, a common biogenic VOC, was not observed during this event, and a brief period of elevated dimethyl sulfide, associated with marine emissions from phytoplankton, was observed shortly before – but not during – this event (Reid et al., 2016).

The data gap on 19 Sept occurred during the return transit to Puerto Princessa, as the trailing winds caused self-sampling of boat emissions. However, westerly winds allowed for sampling once the *Vasco* turned to head into port, and measurements were continued for several hours to characterize emissions from the port and the city with a population of over 200,000. Number concentrations in port were considerably higher than in the remote marine locations of the SCS, with total number concentrations between 4000 and 10,000 cm$^{-3}$ in the 17-500 nm size range. Ultrafine particles ($D_p$ < 100 nm) dominated number concentrations during this period, although large number concentrations of accumulation mode particles with diameters between 100 and 300 nm were also observed.

The *Vasco* departed the port and sailed on an east-southeasterly course late on 20 Sept., during a period when both NOGAPS and the onboard weather tower indicated generally westerly low-level winds. This wind direction allowed for measurements of the Puerto Princessa urban plume as it was transported out over the Sulu Sea. Particle number concentrations between roughly 750 and 2000 cm$^{-3}$ were measured during this period, with size spectra showing a mode with 80-90 nm median diameters. This modal median diameter was unique during the cruise; modal median diameters in the 40-70 nm or 150-225 nm ranges were more commonly observed. As the *Vasco* moved further to the southeast and out of the city plume, the size distribution measurements began to more closely resemble the previously seen bimodal background marine conditions.

The *Vasco* arrived at the remote Tubbataha reef in the middle of the Sulu Sea on 21 Sept and remained there through the end of the measurement period on 27 Sept. Throughout this time, the inflow arm of a nearby tropical cyclone spawned large amounts of intermittent convection and cloud cover over much of the SCS and Borneo. Measured number concentrations and size distributions showed considerable variation during this period as transport of smoke from Borneo was intermixed with cleaner periods associated with precipitation events. A final larger smoke event occurred on 25 and 26 Sept shortly before the end of the measurement period. This event was similar to the early smoke-dominated period with largely mono-modal size distributions and total particle number concentrations above 2000 cm$^{-3}$, and was followed by mixing and a return to background marine conditions.

[revised manuscript text omitted]

~~The hygroscopicity parameter, κ, can be used to quantify the expected role of particle composition on water uptake and activation to cloud droplets (Petters and Kreidenweis, 2007). Anthropogenic pollution from urban areas often includes highly hygroscopic species such as ammonium sulfate (κ = 0.61) and ammonium nitrate (κ = 0.67), although non- or weakly hygroscopic species such as black carbon and nonpolar organic species are also common aerosol components. Fresh sea spray particles, dominated by sodium chloride (κ = 1.28), are expected to have the highest κ values (Good et al., 2010), although co-emitted organic species and replacement of chlorine by uptake of acidic gases can potentially reduce κ. Aged biomass burning aerosol or organic dominated particle populations have generally been found to have κ values below 0.2, while black carbon (κ ≈ 0) has very low hygroscopicity (Andreae and Rosenfeld, 2008; Petters et al., 2009; Engelhart et al.,~~

2012). Similar size-resolved hygroscopicity measurements were performed in a remote rainforest location in Borneo by Irwin et al. (2011) during a time period with little to no biomass burning. They reported κ values between 0.05 and 0.37 for this terrestrial, biogenically dominated MC environment.

The range of κ values measured for the particles active at supersaturations of 0.14% and 0.38% was typically between about 0.3 and 0.8, although the full range was between 0.2 and 1.1 (Figures 2e, f). Average hygroscopicities and standard deviations for each population type at the 0.14% and 0.38% supersaturations are presented in Table 21, along with the average CN and CCN concentrations across all supersaturation settings. It is important to note that particle sizes ($D_{50}$: characteristic particle diameter at which 50% of particles in the CCNc have activated) corresponding to these measurements are in the range of 45 – 150 nm; our measurements did not characterize the hygroscopicities of either the very small particles ($D_{50} < 45$ nm) nor the particles with diameters above ~ 150 nm.

The 0.14% supersaturation scans have $D_{50}$ diameters that span approximately 96 to 150 nm for κ values between the approximate observed range of 0.8 and 0.2, respectively. Hygroscopicity measurements at this lowest supersaturation are therefore more sensitive to particles in the larger accumulation mode – hence our segregation of a subset of observations of κ into accumulation and Aitken parameters for clustering purposes. The averaged properties in Table 21 indicated that such accumulation mode particles had lower average hygroscopicities (κ = 0.40) in the smoke population type as compared to the precipitation, background marine, ultrafine event, and transit populations (κ = 0.54, 0.65, 0.65, 0.58, respectively), while the mixed marine population (κ = 0.48) resided between these.

High concentrations of both $SO_2$ and sulfate aerosol from numerous sources have been observed in the MC (Robinson et al., 2011; Reid et al., 2013). During this study, multi day filter samples showed average sulfate concentrations between approximately 0.8 and 3 μg/m$^3$ at the *Vasco*, potentially increasing during periods of smoke impacts due to burning of sulfur rich peat in the region (Reid et al., 2016). The potential peat source or mixing with other sources of sulfate may explain the higher than typical κ values for aged biomass burning aerosol.

The hygroscopicities derived from measurements at the 0.38% supersaturation set point had different trends. At the 0.38% supersaturation setting, activation occurs for particles sized between roughly 45 and 80 nm for particles in the observed 1.1 to 0.2 κ range, respectively, and thus measurements at this supersaturation were more closely identified with Aitken mode particles. During smoke impacted periods, the Aitken mode particles had κ values of 0.56 as compared with the 0.40 value observed in the accumulation mode. The aged, primary emissions from biomass burning are likely to be confined to particles larger than ~100 nm (Figure 4), and it is therefore possible that the Aitken mode particles in this population were largely derived from background sources rather than from biomass burning, leading to the higher observed κ values.

Interestingly, the opposite situation occurred in the background marine and precipitation aerosol populations, where the Aitken mode was less hygroscopic than the accumulation mode (clean marine: κ = 0.46 and κ = 0.65, respectively; precipitation: κ = 0.34 and κ = 0.54, respectively). Decreasing hygroscopicity with size is consistent with an increasing organic fraction at smaller particle sizes in marine aerosol, as has been observed both in field data and in the laboratory (Cavalli et al., 2004; O'Dowd et al., 2004; Facchini et al., 2008; Prather et al., 2013). These observations are also consistent

with precipitation removal of some of the background sulfate aerosol, leading to lower hygroscopicities in cleaner aerosol populations due to marine organic aerosol becoming more dominant.

Finally, the organic event type had the lowest values (κ ≈ 0.2) in both modes from the entire study. A gas canister grab sample during this period showed elevated levels of a number of organic compounds (Reid et al., 2016; Figure 2e), while the size distributions were similar to those of the background marine population type, but with slightly larger diameters (Figures 3 & 4). These results are consistent with particles dominated by organics across all sizes, perhaps due to growth of a background population by condensation of organics.

**4 Discussion**

[revised manuscript text omitted]
$^3$) | 0.14% SS CCN (#/cm$^3$) | 0.14% SS Act Frac (-) | 0.14% SS $\kappa$ (-) | 0.38% SS CCN (#/cm$^3$) | 0.38% SS Act Frac (-) | 0.38% SS $\kappa$ (-) | 0.53% SS CCN (#/cm$^3$) | 0.53% SS Act Frac (-) | 0.71% SS CCN (#/cm$^3$) | 0.71% SS Act Frac (-) | 0.85% SS CCN (#/cm$^3$) | 0.85% SS Act Frac (-) |
|---|---|---|---|---|---|---|---|---|---|---|---|---|---|---|---|
| 1: Back. Marine | 214 | 231 (111) | 510 (181) | 213 (101) | 0.38 (0.09) | 0.65 (0.11) | 320 (148) | 0.60 (0.12) | 0.46 (0.17) | 416 (194) | 0.74 (0.11) | 444 (239) | 0.81 (0.09) | 480 (210) | 0.87 (0.05) |
| 2: Precipitation | 67 | 142 (79) | 361 (164) | 96 (58) | 0.24 (0.11) | 0.54 (0.14) | 243 (135) | 0.48 (0.15) | 0.34 (0.11) | 352 (175) | 0.65 (0.15) | 265 (82) | 0.71 (0.09) | 228 (100) | 0.79 (0.03) |
| 3: Smoke | 44 | 1800 (273) | 2280 (606) | 1720 (388) | 0.72 (0.04) | 0.40 (0.03) | 2340 (480) | 0.93 (0.02) | 0.56 (0.25) | 1990 (359) | 0.97 (0.02) | 2080 (396) | 0.98 (0.05) | 2150 (523) | 0.99 (0.02) |
| 4: Mixed Marine | 294 | 689 (295) | 975 (271) | 591 (201) | 0.58 (0.08) | 0.48 (0.10) | 827 (270) | 0.83 (0.07) | 0.54 (0.23) | 861 (247) | 0.89 (0.10) | 876 (244) | 0.94 (0.06) | 893 (271) | 0.96 (0.05) |
| 5: Organic Event | 11 | 151 (19) | 277 (30) | 88 (10) | 0.31 (0.02) | 0.22 (0.03) | 144 (9) | 0.53 (0.01) | 0.21 (0.03) | 182 (26) | 0.72 (0.01) | 268 (56) | 0.89 (0.14) | 257 (24) | 0.93 (0.07) |
| 6: Ultrafine Event | 59 | 163 (58) | 577 (158) | 138 (45) | 0.25 (0.06) | 0.65 (0.09) | 361 (172) | 0.56 (0.11) | 0.50 (0.10) | 373 (168) | 0.65 (0.12) | 439 (163) | 0.72 (0.07) | 473 (147) | 0.79 (0.10) |
| 7: Transit | 36 | 311 (44) | 976 (384) | 363 (87) | 0.37 (0.06) | 0.58 (0.08) | 772 (263) | 0.81 (0.09) | 0.62 (0.16) | 832 (423) | 0.87 (0.05) | 877 (370) | 0.90 (0.02) | 878 (195) | 0.95 (0.03) |
| 8: Port | 6 | 671 (210) | 4890 (2550) | 251 (-)* | 0.09 (-)* | 0.49 (-)* | 1126 (-)* | 0.26 (-)* | 0.13 (-)* | 3936 (-)* | 0.40 (-)* | 1742 (289) | 0.57 (0.22) | 2080 (-)* | 0.45 (-)* |
| **All Types** | **731** | **503 (455)** | **851 (677)** | **450 (388)** | **0.47 (0.16)** | **0.54 (0.14)** | **675 (516)** | **0.72 (0.17)** | **0.50 (0.21)** | **698 (555)** | **0.79 (0.15)** | **724 (512)** | **0.85 (0.13)** | **723 (502)** | **0.90 (0.10)** |

* Only one datapoint; Note that Port measurements fluctuated as the Vasco entered port

---

## Author Comment (AC2) · 29 Oct 2016

We would like to thank the reviewer for their insightful comments and helpful suggestions. Each of the comments is listed below in bold text, with our response and corrections following each of them in plain text. We have made a number of small corrections to the manuscript for each of the specific comments as noted. In addition, we have conducted a general reorganization of the results and discussion sections pursuant to the general comments. We have included relevant updated sections in response to general comments, however it may be more effective to reference the marked-up manuscript included at the end of this document to fully address the noted concerns.

**General comments:**

**The paper presents a comprehensive and state-of-the-art aerosol size distribution and CCN dataset taken during a research cruise in the South China Sea. The aerosol cluster analysis in combination with the attribution of aerosol types to air masses and sources is interesting and relevant. The report of representative hygroscopicity values is very useful for the community. The paper is written in a good and clear language, but could be made more concise by dropping repetitions of observations. The scientific content is largely descriptive and should be made stronger by putting the results into context. This is particularly important for the following sections:**

**Conclusions: The meaning of the results should be carved out better and the conclusions made more specific. E.g. p.13, line 31: How exactly do the authors arrive at the conclusions that results are regionally and temporally representative (p. 13, line 31), and what are the further implications? What time period is meant by "temporally" – the season, a whole year? What "previous study"?**

Additional information on comparison of the 2012 study described in this work with the findings of the earlier 2011 pilot study (Reid et al., 2015) was included in the discussion section (included in the response to the discussion section comment) to address the question of the representativeness of these results to the wider SCS remote marine region in the Southwestern Monsoon (SWM) season.

Further, we have added additional components to the conclusion to address the implications of these results. Specifically, we focus on the use of aerosol population properties and their associated variances for providing aerosol microphysical representations appropriate for aerosol microphysical, cloud, and optical modeling of the region. The conclusion section has been updated to:

"5 Conclusion

This study reports ship based measurements of aerosol size distributions and CCN properties conducted as part of the first extensive, in situ aerosol measurement campaign in remote marine regions of the South China Sea/East Sea during the important Southwestern Monsoon and biomass burning season. Analysis of approximately two weeks of measurements found aerosol characteristics consistent with those from a previous pilot study in the region during the same season, indicating that descriptions of aerosol population types and the associated meteorological and transport phenomena that modulate changes and mixing between these populations may be representative of the wider remote marine SCS during the SWM season.

Eight aerosol population types were identified in the dataset that were associated with various impacts from background marine particles, smoke, and anthropogenic sources, as well as precipitation impacts and shorter lived events linked to influxes of VOCs or ultrafine particles. Efforts to measure or model the impact of aerosol on cloud development or atmospheric optical properties often rely on proper characterization of aerosol microphysics associated with impacts from various aerosol sources. As such, we provided population type average values and standard deviations for aerosol size distribution and hygroscopicity properties needed to model aerosol hygroscopic growth in humid environments or cloud development. Future work with this dataset will investigate the impact of the identified aerosol population types on CCN properties including supersaturation dependent CCN concentration needed to model development of different types of clouds. Reutter et al., (2009) identified specific regimes of cloud development where aerosol number concentration was important using a cloud parcel model, while Ward et al., (2010) found such results may be further complicated by aerosol size and hygroscopic properties. Inclusion of both population type average properties and the range that they vary across into such a model may help constrain when various properties of the aerosol are relevant to cloud development in the SCS. Additionally, differences in aerosol population type are expected to be relevant to studies of radiative transfer, optical propagation through the atmosphere, and satellite retrievals in sub-saturated marine environments where differences in particle number concentration, size, hygroscopicity, index of refraction, and relative humidity all affect the interaction radiation with particles in complex ways.

Lastly, while specific observed aerosol population types were identified in this dataset, additional open questions remain regarding the relative importance of various sources and transport pathways of aerosol into remote MBL air masses and their impact on aerosol populations. Since the surface-based observations provide only a portion of the observations needed to construct a true aerosol budget for the MBL, the degree to which MBL aerosol may be impacted by mixing down from a reservoir aloft was not clear. Future airborne aerosol campaigns in the region may be useful to shed light on this important topic."

**Aerosol Hygroscopicity: It would be nice if more of the CCN data and their significance were discussed and if they were put into more context. What supersaturations are expected to be relevant in the regionally typical cloud cover; are the ones chosen here representative? It would be nice if the impact of biomass burning and anthropogenic emissions on all CCN parameters (not just kappa) were carved out a little more.**

In order to provide a more thorough treatment of the CCN properties of the identified population types, the hygroscopicity result section (previously section 3.4) has been removed and its results included directly into the new section 3.2 that discusses each identified cluster type, with parts on background moved to the methods section 2.1. Activated fractions and CCN concentrations across the range of measured supersaturations for population types are now briefly discussed in this section as well. The discussion now has several added paragraphs that place the CCN results in context and explain their significance (see response to the Discussion comment). A sentence was added to section 2.1 regarding choice of supersaturation settings for CCN measurements:

"The measured range of 0.14% to 0.85% supersaturation was selected based on values that could both be reliably measured by the CCNc instrument and represented supersaturations expected in the region where aerosol effects may be relevant, ranging from marine stratocumulus with peak supersaturations often below 0.2% to highly convective clouds with supersaturations above 1% (Reutter et al., 2009; Ward et al., 2010; Tao et al., 2012)."

We also agree that additional treatment of the CCN data and its significance for aerosol-cloud-precipitation interactions in the region is warranted. However, we believe a full treatment of the variability of CCN properties for each population type and their implications in the region requires a more thorough analysis than is feasible within the scope of this manuscript. Here, we intended to focus more on the classification of aerosol population type using the cluster model, and description of the associated aerosol and CCN properties. We have made this point clearer in the conclusion to describe intended future work on CCN properties associated with the identified population types.

**Introduction: The first paragraph (especially up to line 11) should be made more specific. For example, lines 7 and 8: Which additional questions? Representativeness of which results? Also, the first sentence of the introduction is somewhat unfortunate. It begs the question: Why is it important to assess aerosol properties there in the first place? The region and its significance to aerosol research need to be more clearly introduced to the reader.**
We have updated the introduction to provide some additional justification for the study of aerosol properties and specifics as to the open questions regarding its study in the region. The first paragraph now reads:
"In the Southeast Asian Maritime Continent (MC) and South China Sea/East Sea (SCS) aerosol particles are expected to play an important role modulating cloud development, precipitation, and radiative properties that affect heat transfer through the atmosphere (Reid et al., 2013). Assessment of aerosol properties important to understanding such processes in remote marine segments of this region has proven difficult. Extensive cloud cover confounds remote sensing and leads to a clear sky bias in observations (Feng and Christopher, 2013; Reid et al., 2013). Aerosol monitoring has largely been confined to urban centers that are often dominated by local emissions, while in-situ sampling in remote areas has been limited in duration and scope (Irwin et al., 2011; Robinson et al., 2011; Lin et al., 2014; Reid et al., 2015). Airborne measurements have provided some representation of aerosol over wider regions and at various levels (Hewitt et al., 2010; Robinson et al., 2012), but additional questions regarding the representativeness of such point measurements across larger time scales remain. Similarly, the impact of various aerosol sources on surface properties and concentrations in remote marine regions, and their relationship to expected transport pathways and the few remotely sensed column measurements that exist is not well understood. Thus, over these remote ocean regions the aerosol optical and physical properties, their variability in time and space, and the processes controlling aerosol lifecycle have not been well constrained. This uncertainty in the aerosol environment itself comes in addition to uncertainty about its impacts on meteorological processes. Aerosol concentration has been found to relate to cloud development, cloud microphysics, and precipitation formation in the region (Yuan et al., 2011; Wang et al., 2013),

while smoke may affect cloud droplet size distributions and the onset of precipitation, similar to processes observed in other tropical regions impacted by biomass burning (Rosenfeld, 1999; Andreae et al., 2004). Improved knowledge of the aerosol environment and aerosol-cloud-climate relationships in the Southeast Asian region has therefore been identified as important regionally, and in regards to links with global climate and large-scale aerosol budgets (Reid et al., 2013)."

**Discussion: Currently, the section contains mostly repetitions of observations discussed earlier in the paper, and a couple new observations and interpretations. It should be re-worked such that it ties up the results in a way that leads to the conclusions. Alternatively, it could simply be eliminated (the new observations could be discussed in the "results" section).**
The discussion section was intended to provide a coherent description of the nature and causes of changes to the aerosol environment encountered during the cruise, in the context of the previously identified population types. In order to emphasize this conceptual picture of background marine aerosol mixing with other sources and changing due to various phenomena in the discussion, we have first simplified the results section to remove unnecessary repetition and discussion of aerosol measurements.
The results are now presented as a brief overview of the study cruise in section 3.1, followed by a description of the results of the cluster analysis, including hygroscopicity and CCN properties, in section 3.2.
The discussion section is now more clearly constrained to emphasize two points. First, the conceptual model of mixing that follows from the basic clustering results and provides a justification for why the cluster model is appropriate for this analysis is discussed. Second, the CCN results are placed in context against expectations and other reported values.
"4 Discussion
Based on this classification of the SCS remote marine boundary layer aerosol environment, a conceptual picture emerges as to the nature and sources of particles encountered during the Vasco 2012 cruise. A bimodal marine aerosol background was present with number concentrations usually between about 300 and 700 cm-3 and a Hoppel minimum around 90 nm. Primary emissions via sea spray supply submicron particles consisting of a mixture of sea salt and organic components, with emitted particle diameters as small as 20 nm (Clarke et al., 2006; O'Dowd and Leeuw, 2007; Prather et al., 2013). However, even in remote marine environments transported anthropogenic and combustion aerosol may still be an important or even dominant source of small particles (Shank et al., 2012). The background marine population identified in this study is therefore considered a background state across the remote SCS that is likely comprised of a mixture of primary marine emissions along with particles derived from anthropogenic and biomass burning sources throughout the region. Departures from the typical range of background marine characteristics and number concentrations occurred under large influxes of aerosol from other sources, such as smoke from biomass burning regions, anthropogenic pollution from population centers or shipping, or when convection and precipitation removed much of the ambient particulate matter and created relatively clean air masses.
During the SWM when large amounts of biomass burning aerosol were being advected into the SCS, a population of aged, accumulation mode smoke particles was periodically injected into

[revised manuscript text omitted]

**Specific comments:**
**Abstract, line 19: this needs to be re-worded. Right now, the reader might get the impression that the "additional onboard (. . .) model products for the region" somehow entered the cluster analysis.**
We have split this sentence into two in order to clarify this point. It now reads:
"Eight aerosol types were identified using a K-Means cluster analysis with data from a size-resolved CCN characterization system. Interpretation of the clusters was supplemented by additional onboard aerosol and meteorological measurements, satellite, and model products for the region."

**Page 3, lines 16-19 should be moved to section 2.2. What time of day was chosen for the HYSPLIT trajectory arrival? How were the other initial conditions (especially arrival height) motivated?**
We have moved this paragraph to the end of section 2.2 and amended the paragraph as follows:
"The NOAA Hybrid Single Particle Lagrangian Integrated Trajectory (HYSPLIT) Version 4.9 model (Draxler et al., 1999; Draxler and Hess, 1997, 1998) was used to generate daily 72-hour backtrajectories (spawned at 0Z, 8 AM local) from the *Vasco* location with arrival heights of 500m to indicate likely marine boundary layer transport patterns. The GDAS1, 1° x 1° HYSPLIT

meteorological dataset was used to drive the model. Trajectory paths were found to be largely influenced by synoptic scale changes in the regional meteorological state of the atmosphere, with no substantial differences due to arrival time of day. Arrival heights between 100 m and 3000 m were examined. Trajectories with arrival heights below 1000 m were generally consistent and representative of boundary layer transport (Atwood et al., 2013; Xian et al., 2013), while higher heights tended to be increasingly influenced by free troposphere transport pathways with a more westerly component. As such, 500 m was selected to be representative of general shifts in synoptic scale boundary layer transport pathways, though more complex vertical interactions and mixing from aloft are a potential influence in the region (Atwood et al., 2013). Trajectory lengths of 72 hours were found to be sufficient to demonstrate general transport path differences between ocean dominated regions of the central portion of the SCS and more terrestrially influenced regions that passed closer to Borneo and Sumatra."

**Page 3, line 21: What was the height of the mast above the water?**
We have added "approximately 10 m above the water surface" to this sentence.

**Page 3, line 29: What % relative humidity was ensured with this system?**
Occasional in line RH measurements were conducted just downstream of the Nafion dryer that showed RH typically well below 30% RH. We have changed this sentence to read:
"The sample was then dried using a Permapure poly-tube Nafion dryer with low pressure sheath air to RH values below 30%, as verified by occasional in line checks using a handheld Extech Hygro-Thermometer."

**Page 4, line 4 and 8: When and where were the calibrations performed?**
Five calibrations of the CCNc at each supersaturation setting were performed during four sessions throughout the cruise. These occurred near the beginning of the measurement period after recovery from a computer and power supply failure (15 Sep), while anchored at Puerto Princesa (20 Sep), after the last measurements while in transit back to Manila (27 Sep) (during which two full calibrations were conducted), and after arrival in Manila (29 Sep). Each session took two to five hours to complete, hence only four were conducted to limit measurement downtime. All CCNc supersaturation calibration discussion was moved to the methodology in section 2.1, as noted in the next comment.
CCN system flow rates were calculated with an in-line Gilibrator just before each CCNc supersaturation calibration session. The sentence describing this process has been amended to read:
"Flow rates used to calculate number concentrations were calibrated using a Gilibrator (Models 800285 & 800286) system, with in-line measurements conducted prior to each CCNc supersaturation calibration session as noted below."

**Page 4, line 10: Can you comment on the stability of the supersaturation settings throughout the cruise?**
No significant trend was noted in the calibrations, which are noted to have supersaturation dependent calibration standard deviations of "0.14% ± 0.01, 0.38% ± 0.01, 0.52% ± 0.01, 0.71% ± 0.02, and 0.85% ± 0.03." As noted in the previous comment, these are based on five

calibrations, as they took approximately two to five hours to complete. All supersaturation calibration discussion was moved to section 2.1 and now reads:

"Full calibration of the CCN system flow rates and supersaturations took two to five hours to complete, and was therefore conducted four times throughout the study on 15, 20, 27 and 29 September to limit measurement downtime. Each calibration session involved running a calibration scan at each CCNc temperature gradient setting (with two full scans conducted at each setting on 27 Sep) yielding a total of five calibrations per setting throughout the cruise. Calibrated supersaturation set-points and their respective standard deviations were 0.14% ± 0.01, 0.38% ± 0.01, 0.52% ± 0.01, 0.71% ± 0.02, and 0.85% ± 0.03, with no significant trend or calibration drift noted during the cruise."

**Page 4, line 25: What was the temporal resolution of the filter samples? Are the analysis methods described in more detail in Reid et al., 2016? If so, the reference should be added to this sentence, too.**

Filter sample periods were between one and two and half days, and are described further by Reid et al., 2016. The sentence has been changed to:

"A series of $PM_{2.5}$ filters were collected by 5 lpm Minivol Tactical Air Samplers with sampling periods that varied between one and two and half days, and were analyzed for elemental concentrations by gravimetric, XRF, and ion chromatography methods, and organic and black carbon concentrations by the thermal-optical methods (Reid et al., 2016)."

**Page 5: please state the specific products used in this study (for example, where does the AOD mentioned in page 6, line 13, come from)?**

We have updated this section to include this information. The following sentence was added: "The MODIS Collection 6 MOD08 Level 3 daily Aerosol Optical Depth products were utilized for AOD measurements in the region, though cloud cover obscured measurements throughout much of the study."

**Page 6, line 8: How were "surface winds" averaged, over what period or area?**

This information was added to the earlier initial description of the use of NOGAPS data in the methods section 2.2. The sentence now reads:

"Simulations from the Navy NOGAPS model were used to represent surface and 700 hPa winds, interpolated to one-degree spatial resolution and averaged over the study period, to provide an estimate of typical aerosol transport pathways (Hogan and Rosmond, 1991; Xian et al., 2013)."

**Page 6, line 16: Why was only this one density used - fire emissions were presumably not the only coarse aerosol type in the region?**

The purpose of this initial reconstruction was only to provide a general estimate of which periods had generally larger mass concentrations in order to provide a broad indication of how aerosol impacts changed throughout the study. Unit density or volume could provide similar indications, but the largest plumes associated with the largest mass concentrations were identified as smoke in an earlier study (Reid et al., 2016). As such, we opted to use a density consistent with smoke dominated aerosol to provide somewhat more realistic comparisons to NAAPS results. Overall, this section is only intended to provide an initial justification for the

reader that different aerosol population impacts are to be expected throughout the study. We have modified this section to indicate these are merely estimates as follows:
"Accumulation mode aerosol mass concentration estimates (Figure 1b) were derived from the PCASP measurements using a density of 1.4 µg m$^{-3}$ (Levin et al., 2010), assumed to be representative of a combination of smoldering peat and agricultural fire emissions typical in the MC (Reid et al., 2012) that constituted the largest plumes observed the study (Reid et al., 2016). Coincident model estimates generated by NAAPS along the ship track indicated generally similar results, with the highest mass concentrations occurring early and late in the measurement period, in general agreement with times during which backtrajectories passed over terrestrial sources and active fires".

**Page 6, line 28-29: It would be helpful to label these locations in the map in Figure 1.**
Location names and anchorage times have been added to Figure 1. A copy of the updated figure along with additional changes can be found later in this document in a separate comment on Figure 1.

**Page 7, line 3: perhaps specify "biomass burning smoke". Are there any filter analyses for this period?**
This change has been made in the new results section 3.2 section on the smoke type. The sentence has been clarified to discuss measurements used to attribute the measured aerosol to biomass burning smoke, and an additional sentence has been added indicating more attribution of this event to biomass burning was conducted in Reid et al., (2016). The section now reads:
"Accumulation mode lognormal median diameters around 200 nm with a tail of smaller particles, elevated concentrations of carbon monoxide and benzene, as well as potassium in filter samples during this period (Reid et al., 2016, and Figure 2) were all consistent with expectations for aged biomass burning smoke (Yokelson et al., 2008; Akagi et al., 2011; Reid et al., 2015; Sakamoto et al., 2015). Additional examination and attribution of this event to biomass burning in Sumatra and Borneo is discussed further in Reid et al., (2016)."

**Page 9, line 7: it would be nice to see the tri-modal fit included in Figure 4**
An additional figure has been added as the new supplementary figure S1 that compares the two and three mode fits. The discussion of the ultrafine event type in the results section now references this figure as well.

[Figure]

**Supplemental Figure S1:** The best fit of the Normalized dN/dlogD$_p$ particle size distributions for the Ultrafine Event population type using two lognormal modes (left) and three lognormal modes (right). A three mode fit was indicated by the Hussein et al., (2005) algorithm for this population type, however for clustering purposes the two mode fit was used. Shown in these plots are the particle size distribution spectrum for each data point within the population (grey), the associated average bin values with error bars at 95% confidence interval (colored step lines; 1.96 * bin standard deviation), and best fit lognormal modes (colored curves) with multi-modal fit (black). The average particle number concentrations of data points within each population type are listed.

**Page 10, lines 23 – 33: This paragraph should be moved into the introduction. If a comparison of literature kappa values to this study's was intended, this comparison should be done more directly, rather than expecting the reader to jump back and forth between paragraphs.**

The background description of Kappa was intended primarily as a primer for the reader regarding interpretation of the hygroscopicity results. We have moved the literature comparison directly to the discussion section as noted in the response to the general discussion comment.

**Page 11, lines 23: -29: How can it be stated that the Aitken mode during the biomass burning period be derived from the background aerosol, when it is actually more hygroscopic than the Aitken mode of the marine background and precipitation clusters? Why would the biomass influence be confined to particles >100nm? (Surely not all biomass burning particles are primary?)**

It was not our intention to state the origin of Aitken size range particles in the tail of the distribution during the biomass burning smoke periods, for which there is not enough information to state with certainty. Rather, we intended to note that the difference in kappa values between Aitken and accumulation mode particles moved in opposing directions for smoke and marine population types. The increased kappa values for Aitken mode particles may be due to some combination of background marine Aitken particles, condensation or secondary formation of Aitken mode aerosol from higher hygroscopicity gas phase species co-emitted with the biomass burning, or similar effects from urban or anthropogenically influenced emissions along the transport pathway. Additionally, the Aitken mode hygroscopicity measurements were subject to higher uncertainties and varied more than the accumulation mode measurements.

We agree that the treatment of the hygroscopicity could therefore be clarified in the manuscript. We have removed much of the discussion of the possible causes of differences in hygroscopicity between population types from section 3.4 and re-focused the results section on presenting the population results. The discussion on hygroscopicity in section 4 has now been better constrained to note these points.

The updated discussion section has been discussed and copied in the earlier responses to the general comments. The updated results can be found in the marked-up version of the manuscript included at the end of this document.

**Language/typos:**
**Abstract, line 20: "aerosol population that" should be "aerosol population and"**
This change has been made.

**Page 2, line 31: "in situ" is not the right expression here, I believe. "In the area" or "in the SCS" would be better.**
We used in situ in this case primarily to contrast the measurements with remote sensing observations noted earlier in the introduction, which are both difficult to conduct and insufficient to properly characterize the region's aerosol. We therefore would prefer to keep the "in situ" term in this sentence, but we have added "in the area" to the sentence to clarify this point. It now reads:
"These measurements represent the first in situ observations of size-resolved CCN properties in the area, and fill a gap in knowledge needed to assess aerosol-cloud-precipitation relationships in the in the data poor remote marine SCS region."

**Page 4, line 15 and 16: choose one, "CCN activation spectrum" or "activated fraction spectrum"**
We have clarified these two terms as well as the quantities being measured and calculated to remove the confusing usage. This section has been rewritten as follows:
"The inversion yielded the dry ambient aerosol size distribution over the measured range ($dN/dlog_{10}Dp$ for $17 \leq Dp \leq 500$ nm) and the equivalent distribution of CCN particles activated at each supersaturation ($dCCN/dlog_{10}Dp$). The activated fraction spectrum was then calculated as the fraction of total particles that formed droplets (CCN/CN) at each diameter Dp. Each activated fraction spectrum was then fit using a three parameter fit similar to the approach of Rose et al. (2010)."

**Page 4, line 25: "that were analyzed" should be "and were analyzed".**
This change has been made and sentence has been changed to:
"A series of PM2.5 filters were collected by 5 lpm Minivol Tactical Air Samplers with sampling periods that varied between one and two and half days, and were analyzed for elemental concentrations by gravimetric, XRF, and ion chromatography methods, and organic and black carbon concentrations by the thermal-optical methods (Reid et al., 2016)."

**Page 4, line 14: A new sentence would be better than the hyphenation-clause combination:
"number concentration). Two modes were identified as the best fit. . ."**
This change has been made.

**Page 4, line 16 and line 25: Start a new sentence instead of the hyphenation.**
These changes have been made.

**Page 4, line 22: "shortest" instead of "closest"?**
This does provide a better description. We have made the change.

**Page 8, line 31: Start a new sentence after "(Figure 2d)".**
This change has been made.

**Page 11, line 12: "lower (. . ..) than in the precipitation (. . .) populations"**
This section has been removed and the information moved to the results and discussion
sections, and reworded to no longer include this sentence.

**Page 12, line 1: What does "from the entire study" mean?**
Hygroscopicity measurements during the organic event type were among the lowest measured
during this study. This information has been moved to the new results section 3.2 for the
organic event and now reads:
"Both Aitken and accumulation mode particles had among the lowest hygroscopicities
measured during the cruise, with $\kappa$ values around 0.2."

**Figures/Tables:**
**Caption of figure 4: "each spectrum"**
This change has been made.

**Figure 1: The tick marks on the time axis would be easier to identify if they pointed outward**
We have changed Figure 1 based on comments to the following.

[Figure]

**Table 1: The font size is too small.**
We have included a rotated table in landscape orientation on the page, and will work with the publisher to ensure the new tables are better formatted for publication. If needed, we will split the table into several lines.

[revised manuscript text omitted]

**3.2 Analysis of Daily Observations**

At the start of the measurement period on 14 Sept., the *Vasco* was in transit south from Puerto Princessa, Philippines, in the middle of the island towards Balabac Island at the southern edge of Palawan aArchipelago. HYSPLIT 500 m backtrajectories on this day identified MBL transport from the southwest over burning regions in Borneo (Figure 1a). As shown in Figure 2, at the start of the campaign, high number concentrations of accumulation mode particles (1500 to 4000 cm$^{-3}$) with generally mono-modal size distributions were observed (best-fit lognormal median diameters around 200 nm), consistent with expectations for aged smoke. Elevated concentrations of potassium, carbon monoxide, and benzene during this period (Reid et al., 2016, and Figure 2c) all reinforce the classification of this aerosol as smoke (Yokelson et al., 2008; Akagi et al., 2011; Reid et al., 2015). This initial smoke impacted period was followed by a drop in particle number concentrations after 12Z on 14 Sept. as a squall line passed over the *Vasco* and left a clean air mass scrubbed of many of the particles. Total particle number concentrations were among the lowest values observed during the cruise, with fewer than 200 cm$^{-3}$ measured in the 17-500 nm range. Particle number concentrations did not recover for approximately 10 hours, at which point the aerosol size spectra and concentrations resembled those measured in the air mass before the gust front passage.

The *Vasco* remained at the same anchorage at Balabac Island on the southern tip of the Palawan Archipelago until 19 Sept. Beginning at about 5Z on 17 Sept., particle number concentrations dropped, but not as low nor as rapidly as during the post-squall line event. A distinct bimodal size distribution was observed that coincided with the onset of precipitation observed at the boat (Figure 2d). Satellite visible and IR imagery (not shown) indicated that no long-lived squall line was associated with this 17 Sept. precipitation event. Beginning around 18Z on 17 Sept. number concentrations began to increase and a third

mode of particles with diameters between 20 and 30 nm was observed (Figure 2a). This mode slowly mixed into the Aitken mode as more typical bimodal distributions resumed by the middle of 18 Sept. A filter during this period showed very low potassium concentrations, with benzene among the lowest values measured during the study, indicating that biomass burning was not the likely source for this event. In addition, at the start of this period several volatile organic compound species (VOCs) were elevated as compared to periods before and after the event. Anthropogenic, shipping, and marine and terrestrial biogenic emissions are known sources of such compounds; isoprene, a common biogenic VOC, was not observed during this event, and a brief period of elevated dimethyl sulfide, associated with marine emissions from phytoplankton, was observed shortly before – but not during – this event (Reid et al., 2016).

The data gap on 19 Sept occurred during the return transit to Puerto Princessa, as the trailing winds caused self-sampling of boat emissions. However, westerly winds allowed for sampling once the *Vasco* turned to head into port, and measurements were continued for several hours to characterize emissions from the port and the city with a population of over 200,000. Number concentrations in port were considerably higher than in the remote marine locations of the SCS, with total number concentrations between 4000 and 10,000 cm$^{-3}$ in the 17-500 nm size range. Ultrafine particles ($D_p$ < 100 nm) dominated number concentrations during this period, although large number concentrations of accumulation mode particles with diameters between 100 and 300 nm were also observed.

The *Vasco* departed the port and sailed on an east-southeasterly course late on 20 Sept., during a period when both NOGAPS and the onboard weather tower indicated generally westerly low-level winds. This wind direction allowed for measurements of the Puerto Princessa urban plume as it was transported out over the Sulu Sea. Particle number concentrations between roughly 750 and 2000 cm$^{-3}$ were measured during this period, with size spectra showing a mode with 80-90 nm median diameters. This modal median diameter was unique during the cruise; modal median diameters in the 40-70 nm or 150-225 nm ranges were more commonly observed. As the *Vasco* moved further to the southeast and out of the city plume, the size distribution measurements began to more closely resemble the previously seen bimodal background marine conditions.

The *Vasco* arrived at the remote Tubbataha reef in the middle of the Sulu Sea on 21 Sept and remained there through the end of the measurement period on 27 Sept. Throughout this time, the inflow arm of a nearby tropical cyclone spawned large amounts of intermittent convection and cloud cover over much of the SCS and Borneo. Measured number concentrations and size distributions showed considerable variation during this period as transport of smoke from Borneo was intermixed with cleaner periods associated with precipitation events. A final larger smoke event occurred on 25 and 26 Sept shortly before the end of the measurement period. This event was similar to the early smoke dominated period with largely mono-modal size distributions and total particle number concentrations above 2000 cm$^{-3}$, and was followed by mixing and a return to background marine conditions.

**3.3 2 Aerosol Population Type Classification and Properties**

The cluster analysis was first conducted to investigate potential aerosol population types in the dataset, followed by physical interpretation of the results against cluster aerosol properties, coincident measurements, and meteorological conditions. The parameter values input to the cluster analysis are shown for each data point and variable in Figure 3, and colored by cluster number for the results of the eight-cluster K-Means analysis. The average value and intra-cluster standard deviation for each cluster parameter and cluster are given in Table 1. Normalized size distributions for each of these eight aerosol populations are shown in Figure 4; the average CN and CCN number concentrations and hygroscopicities are given in Table 2 1. Equivalent normalized volume distributions are shown in Supplementary Figure S1S2. The cluster number associated with each measurement is similarly shown as the background color in Figure 2 and marker color in Figure 5. The aerosol properties, meteorological conditions, and likely transport pathways associated with data points in each cluster were then used to provide a physical interpretation of the results and A name identifying the likely source of each population type on the basis of its likely sources as discussed below. was then assigned as follows, on the basis of the previously identified meteorological and other factors discussed in Sect. 3.2. Clusters 1-4 were the most commonly found (representing 85% of the total observations, Table 2 1), while clusters 5-8 represented special cases, generally of short duration, that could be identified with by specific locations or sampling conditions.

1. Background Marine: Data points associated with this cluster occurred throughout the study, typically following rain in the vicinity of the *Vasco* or transport from areas further removed from terrestrial regions. In addition, this type was observed following shortly after periods associated with each of the other identified clusters, often appearing as a transition between other types (Figure 2). The measured properties of T this population type was were similar to the background marine aerosol reported in many prior studies (e.g. Hoppel et al., 1994; O'Dowd et al., 1997; Spracklen et al., 2007; Good et al., 2010). The population featured a , and consisted of a bimodal size distribution with a Hoppel minimum near 90 nm, thought to be due to cloud or fog processing of marine aerosol (Hoppel et al., 1986) near 90 nm due to cloud or fog processing. The inner quartile range (IQR: middle 50% of observations between the 25%–75% percentiles) of number concentrations ranged from 382 to 623 cm$^{-3}$, with on average 42% of the total number concentration residing in the accumulation mode as specified by the bimodal fit. Modal hygroscopicities were found to average 0.65 for the accumulation mode and 0.46 for the smaller Aitken mode, while activated fractions were generally moderate across the range of measured supersaturations as compared to other identified population types (Table 1). Each of these findings further reinforced the classification of this cluster as a typical background marine aerosol.

2. Precipitation: This distribution was found during periods immediately following extensive precipitation at or near the *Vasco* (Figure 2d). A in scrubbed air masses had been substantially scrubbed of particles and where accumulation mode particles had been preferentially removed. While the number concentrations of large-mode particles were lower than those in the background marine periods, the number concentrations of smaller particles, particularly

those below 40-50 nm, were comparable to the  background marine type. The longest contiguous period of this type occurred on 14 Sept. immediately following the passage of a squall line observed in the satellite visible and IR products (not shown) that left a clean air mass with fewer than 200 cm$^{-3}$ measured in the 17-500 nm range in its wake. and Extended periods of this type occurred in the wake of the squall lines, though not all instances of nearby precipitation lead to this type. 
[revised manuscript text omitted]

~~The hygroscopicity parameter, κ, can be used to quantify the expected role of particle composition on water uptake and activation to cloud droplets (Petters and Kreidenweis, 2007). Anthropogenic pollution from urban areas often includes highly hygroscopic species such as ammonium sulfate (κ = 0.61) and ammonium nitrate (κ = 0.67), although non- or weakly hygroscopic species such as black carbon and nonpolar organic species are also common aerosol components. Fresh sea spray particles, dominated by sodium chloride (κ = 1.28), are expected to have the highest κ values (Good et al., 2010), although co-emitted organic species and replacement of chlorine by uptake of acidic gases can potentially reduce κ. Aged biomass burning aerosol or organic dominated particle populations have generally been found to have κ values below 0.2, while black carbon (κ ≈ 0) has very low hygroscopicity (Andreae and Rosenfeld, 2008; Petters et al., 2009; Engelhart et al.,~~

2012). Similar size-resolved hygroscopicity measurements were performed in a remote rainforest location in Borneo by Irwin et al. (2011) during a time period with little to no biomass burning. They reported $\kappa$ values between 0.05 and 0.37 for this terrestrial, biogenically dominated MC environment.

The range of $\kappa$ values measured for the particles active at supersaturations of 0.14% and 0.38% was typically between about 0.3 and 0.8, although the full range was between 0.2 and 1.1 (Figures 2e, f). Average hygroscopicities and standard deviations for each population type at the 0.14% and 0.38% supersaturations are presented in Table 21, along with the average CN and CCN concentrations across all supersaturation settings. It is important to note that particle sizes ($D_{50}$: characteristic particle diameter at which 50% of particles in the CCNc have activated) corresponding to these measurements are in the range of 45 – 150 nm; our measurements did not characterize the hygroscopicities of either the very small particles ($D_{50}$ < 45 nm) nor the particles with diameters above ~150 nm.

The 0.14% supersaturation scans have $D_{50}$ diameters that span approximately 96 to 150 nm for $\kappa$ values between the approximate observed range of 0.8 and 0.2, respectively. Hygroscopicity measurements at this lowest supersaturation are therefore more sensitive to particles in the larger accumulation mode – hence our segregation of a subset of observations of $\kappa$ into accumulation and Aitken parameters for clustering purposes. The averaged properties in Table 21 indicated that such accumulation mode particles had lower average hygroscopicities ($\kappa$ = 0.40) in the smoke population type as compared to the precipitation, background marine, ultrafine event, and transit populations ($\kappa$ = 0.54, 0.65, 0.65, 0.58, respectively), while the mixed marine population ($\kappa$ = 0.48) resided between these.

High concentrations of both $SO_2$ and sulfate aerosol from numerous sources have been observed in the MC (Robinson et al., 2011; Reid et al., 2013). During this study, multi-day filter samples showed average sulfate concentrations between approximately 0.8 and 3 μg/m$^3$ at the *Vasco*, potentially increasing during periods of smoke impacts due to burning of sulfur rich peat in the region (Reid et al., 2016). The potential peat source or mixing with other sources of sulfate may explain the higher than typical $\kappa$ values for aged biomass burning aerosol.

The hygroscopicities derived from measurements at the 0.38% supersaturation set point had different trends. At the 0.38% supersaturation setting, activation occurs for particles sized between roughly 45 and 80 nm for particles in the observed 1.1 to 0.2 $\kappa$ range, respectively, and thus measurements at this supersaturation were more closely identified with Aitken mode particles. During smoke impacted periods, the Aitken mode particles had $\kappa$ values of 0.56 as compared with the 0.40 value observed in the accumulation mode. The aged, primary emissions from biomass burning are likely to be confined to particles larger than ~100 nm (Figure 4), and it is therefore possible that the Aitken mode particles in this population were largely derived from background sources rather than from biomass burning, leading to the higher observed $\kappa$ values.

Interestingly, the opposite situation occurred in the background marine and precipitation aerosol populations, where the Aitken mode was less hygroscopic than the accumulation mode (clean marine: $\kappa$ = 0.46 and $\kappa$ = 0.65, respectively; precipitation: $\kappa$ = 0.34 and $\kappa$ = 0.54, respectively). Decreasing hygroscopicity with size is consistent with an increasing organic fraction at smaller particle sizes in marine aerosol, as has been observed both in field data and in the laboratory (Cavalli et al., 2004; O'Dowd et al., 2004; Facchini et al., 2008; Prather et al., 2013). These observations are also consistent

~~Finally, the organic event type had the lowest values ($\kappa \approx 0.2$) in both modes from the entire study. A gas canister grab sample during this period showed elevated levels of a number of organic compounds (Reid et al., 2016; Figure 2e), while the size distributions were similar to those of the background marine population type, but with slightly larger diameters (Figures 3 & 4). These results are consistent with particles dominated by organics across all sizes, perhaps due to growth of a background population by condensation of organics.~~

**4 Discussion**

[revised manuscript text omitted]

---

## Author Response (AR2)

Response to Editor:

15

20

We would like to thank the Editor for the helpful comments and have updated the manuscript to reflect changes based on these remarks. We have included the comments in **bold** and our responses in plain text following each comment. At the end

5 of our response, we have also included a marked up version of the manuscript that identifies changes since the last response. Following this new marked up version is the previous response and markup document.

**Non-public comments to the Author:**

- 10 To assist you in responding to reviews, I note there are several key references on marine aerosols that were not discussed (including those from your own coauthors...) in your original manuscript or in the responses, that seem relevant to the issues raised by the reviewers.
  - marine kappa -- there are several papers by Quinn (e.g. Quinn et al., 2014, Nature Geoscience) that to me appear more directly relevant, plus isn't there also a lab study by Markus (and you) on marine kappa? Keene et al. also is likely a better reference for size dependent organic fraction (by mass).

Additional background information on marine kappa and hygroscopicity measurements have been included in the discussion of our marine kappa results. In particular, to address this comment and the related comments by Reviewer 1 regarding salt and organics, we have added findings from several other sources including those that are noted, and used them to assist with interpretation of our results. The updated paragraph now reads:

- "While the spectrum of mixing between population types is relevant to the identification of impacts from various sources, additional consideration of these aerosol types against measurements in other regions is also warranted. Fresh sea spray particles, dominated by sodium chloride ( $\kappa = 1.28$ ), are expected to have the highest  $\kappa$  values, although co-emitted organic species and replacement of chlorine by uptake of acidic gases can potentially reduce hygroscopicities. Additionally, the
- 25 increasing organic fractions at smaller sizes reported in sea spray aerosol (Keene et al., 2007; Prather et al., 2013; Quinn et al., 2014; Forestieri et al., 2016) lead to decreased hygroscopicities as particle size decreases. Reported hygroscopicities for aerosol in marine regions vary, with Good et al., (2010) reporting CCN-derived κ values above 1 for some background or marine dominated MBL air masses, consistent with pure sodium chloride dominated sea salt κ values. Prather et al., (2013) generated aerosol using sea water with varying organic concentrations and associated marine biological activity, and
- 30 reported CCN activation diameters at 0.2% supersaturation that correspond to  $\kappa$  values above 1 during periods of low organic concentration, that then dropped to  $\kappa$  values as low as 0.1 as organic concentration was increased. In addition, they found more organic enrichment and lower associated hygroscopicities in smaller particles. Quinn et al., (2014) explored the relationship between organic aerosol content, particle size, and particle hygroscopicity in primary marine aerosol generated in several ocean regions. They found a similar enrichment of average aerosol organic volume fraction with decreasing

particle size (40 nm dry diameter: 0.8 organic volume fraction; to 100 nm: 0.4 fraction) that corresponded with decreased hygroscopicities (40 nm:  $\kappa$ =0.4; 100 nm:  $\kappa$ =0.8), and was consistent in various ocean regions and largely independent of biological activity as indicated by chlorophyll-a levels. Such findings are consistent with the lower Aitken mode hygroscopicities found in the background marine populations observed in this dataset, as well as the additional decreases in

5 hygroscopicity noted in the precipitation population that had been further scrubbed of larger accumulation mode particles. Our findings of background and precipitation impacted marine aerosol κ values generally between 0.2 and 0.6 for the Aitken mode and 0.4 and 1.0 for the accumulation mode are therefore consistent with reported hygroscopicities of background marine aerosol that had been enriched by organic components. The presence of organics explains the noted lower hygroscopicities, as compared to what would be expected from pure sea salt aerosol, in population types that were otherwise

10 expected to be dominated by background marine aerosol."

15

**2) clustering to separate marine/nonmarine -- seems very similar to what has been done by other authors.**

We have included additional discussion and references to other studies that utilized clustering methods to discriminate between aerosol types based on both size distribution and composition in our methods section. The updated paragraph now reads:

"Parameters associated with each data point were then used to cluster the data points into groups based on similar observed aerosol properties. Cluster analyses have long been used to group observed aerosol size distributions into clusters of generally similar size distributions (Tunved et al., 2004), which can then be associated with various sources or atmospheric processes that shaped them (Charron et al., 2008; Beddows et al., 2009; Wegner et al., 2012). Similar cluster analyses have

- 20 been utilized to classify aerosol types based on particle chemistry (Frossard et al., 2014), with Frossard et al., (2014) identifying clusters in marine aerosol observations associated with marine and non-marine aerosol types. As pointed out by those authors, the clustering approach can be superior to algorithms using simpler criteria to distinguish "clean" from "polluted" conditions, as more variables and a measure of similarity between data points are used to find the underlying population types. In this study normalized size distribution parameters were combined with total number concentration and
- 25 particle composition information via hygroscopicity measurements to serve as input variables for the cluster analysis (parameters given in Table 1). As hygroscopicity data were available for at most one mode (during the 0.14% or 0.38% supersaturation scans), Aitken and accumulation mode hygroscopicities were treated as missing for data points without this information. In order to account for missing data and adjust all clustering variables to the same scale, each variable was first standardized to a mean of zero and standard deviation of one, with missing data points imputed to a value a zero (the mean
- 30 value). As a result, the clustering distance function was insensitive to missing data, but still included information on hygroscopicity when available."

**3) marine size distributions and wind speed dependence is reported at length by several papers. Your approach here is different but seems to me consistent and should be compared.**

The relationship between wind speed and marine size distributions is now more fully treated in two locations. It is covered in a more thorough discussion of marine aerosol in the introduction (see response to next comment), and in the results section where we further address Reviewer 1's question regarding if the impact of wind speed could be observed in the cluster results or merely in the coarse mode number concentrations. The two paragraphs in the results section addressing this now read:

5

"Finally, throughout the study coarse mode particles with diameters larger than about 800 nm were consistently observed in the PCASP volume distributions (Figure 2b). Concentrations of particles in this size range increased with increasing wind speed (Figure 5), consistent with generation of sea spray aerosol due to bubble breaking and wave action (O'Dowd and Leeuw, 2007). While the total number concentration of coarse particles is small compared to typical CCN concentrations

- 10 (Figures 2e, f), in the cleanest conditions we measured, they represented non-trivial fractions of CCN active at 0.14% and 0.38% supersaturations. The large diameter of these particles makes them likely to activate at very low supersaturations, and they are present in more than sufficient number concentration to impact the microphysical structure and processes in stratocumulus clouds by serving as "giant CCN" (Feingold et al., 1999).
- No significant relationship between wind speed and fine mode aerosol population type was noted. However, particles in the coarse mode range are not measured or accounted for in our cluster analysis (CCN system range: 17-500 nm), while submicron aerosol was often dominated by aerosol from other sources. Modini et al., (2015) utilized a dedicated size distribution fitting analysis that included size resolved observations of particles above 500 nm to examine primary submicron marine aerosol production. They found a primary mode with a median diameter around 200 nm and tail that extended to sizes well above 500 nm, with number concentrations of 12 +/- 2 cm-3 during a period of low wind speeds that increased to
- 20 71 +/- 2 cm-3 as winds increased. Concentrations differences of around 50-60 cm-3 due to wind speed changes may not have resulted in large enough changes to concentrations or size distributions to alter the clustering of the observed population types, which often included number concentrations that were larger by an order of magnitude or more."
- 4) Part of the introduction also seems odd; it almost sounds like the justification for measuring remote aerosol 25 from ship is needed because it has never been done before...or at least not since Hoppel. Several dozen marine aerosol cruises by Quinn, Bates, Shantz, Leaitch, seem relevant. We have added an additional paragraph to the introduction that provides a better overview of previous studies and current understanding of marine aerosol and the sources and processes that impact it. The last two paragraphs of the introduction now read:
- 30 "Remote marine aerosol and its impact on atmospheric processes have been studied in a number of ocean regions (Hoppel et al., 1986; Russell et al., 1994; Jensen et al., 1996; Brechtel et al., 1998; Murphy et al., 1998; Bates et al., 2000; Petters et al., 2006; Quinn et al., 2006). These studies identified a background submicron marine aerosol that is composed of two distinct modes in the number distribution, due to processing by non-precipitating clouds (Hoppel et al., 1986, 1994; Hudson et al., 2015). Bates et al., (2000) linked the differences in the average size distributions of background marine aerosol in two

remote marine regions to regional meteorology, including differences in aerosol residence time and cloud processing. Increased wind speeds lead to increased flux of sea-salt particles into the atmosphere, contributing submicron particles as small as 40 nm in diameter (O'Dowd and Leeuw, 2007; Russell et al., 2010; de Leeuw et al., 2011; Bates et al., 2012; Modini et al., 2015). Non-seasalt-sulfate and organic matter from marine sources also comprise large fractions of the

- 5 submicron aerosol mass loading in clean and background marine air masses (Murphy et al., 1998; Cavalli et al., 2004). As air masses from more terrestrial or anthropogenically-influenced regions advect over remote marine regions, submicron size distributions and chemical compositions often diverge from background conditions (Bates et al., 2000; Quinn et al., 2006). More recent studies have further quantified the role of various processes in shaping the marine aerosol population, including primary and secondary production, aging, and mixing with non-marine sources (Allan et al., 2009; Russell et al., 2010; de
- 10 Leeuw et al., 2011; Bates et al., 2012; Prather et al., 2013; Frossard et al., 2014; Modini et al., 2015). In particular, the contribution of dissolved organic components in the sea surface micro layer to aerosol produced by bubble breaking has been noted, with increasing organic enrichment as size decreases (Russell et al., 2010; Bates et al., 2012; Prather et al., 2013; Quinn et al., 2014). Additional studies into the source dependent composition of marine aerosol have indicated non-marine sources can be important contributors to aerosol in marine regions. Shank et al., (2012) found evidence of biomass burning
- 15 and combustion impacts on remote marine MBL aerosol, including in nominally clean marine conditions. These authors also noted the limited importance of organic components in particulate matter in a tropical Pacific location, as compared to other regions where organics were a more important fraction of the submicron aerosol. Frossard et al., (2014) found influences on aerosol organic matter from shipping and mixing with non-marine sources in 63% of observations across five ocean regions. Modini et al., (2015) evaluated the contribution of primary marine aerosol to cloud condensation nuclei (CCN) number
- 20 concentrations, and found that it accounted for less than 10% of CCN active at 0.9% supersaturation during low wind conditions, with increasing importance (up to 58% of CCN) at higher wind speeds and lower environmental supersaturations. Taken as a whole, recent understanding of marine aerosol indicates that the background marine aerosol and primary marine emissions can be complex and play an important role in cloud, radiative, and precipitation processes, and that other sources of aerosol contribute to number and mass concentrations, even in relatively clean and/or remote regions.
- 25 Two research cruises were conducted in the remote marine boundary layer (MBL) of the SCS and Sulu Sea during the 2011 and 2012 SWM seasons to perform in situ aerosol and meteorological measurements, and to investigate marine aerosol and its impacts on clouds, precipitation, and climate as it reflects the complex set of sources in the region (Reid et al., 2015, 2016). In this paper, we present observations of aerosol and CCN characteristics during the second cruise, along with their relationship to aerosol source type, air mass, and meteorological phenomena. These measurements represent the first in situ
- 30 observations of size-resolved CCN properties in the area, and fill a gap in knowledge needed to assess aerosol-cloudprecipitation relationships in the data poor remote marine SCS region."

Authors' marked up changes since the last response:

**Size-resolved aerosol and cloud condensation nuclei (CCN) properties in the remote marine South China Sea, Part 1: Observations and source classification**

Samuel A. Atwood1, Jeffrey S. Reid2, Sonia M. Kreidenweis1, Donald R. Blake3, Haflidi H. Jonsson4, 5 Nofel D. Lagrosas5, Peng Xian6, Elizabeth A. Reid2, Walter R. Sessions6,7, James B. Simpas5

1Department of Atmospheric Science, Colorado State University, Ft. Collins, CO
 2Marine Meteorology Division, Naval Research Laboratory, Monterey, CA
 3Department of Chemistry, University of California, Irvine, CA
 4Department of Meteorology, Naval Postgraduate School, Monterey, CA
 5Manila Observatory, Manila, Philippines
 6CSC Inc. at Naval Research Laboratory, Monterey, CA

[revised manuscript text omitted]